# Structure and Characterization of Vacuum Arc Deposited Carbon Films—A Critical Overview

**Bernd Schultrich**

Fraunhofer Institute for Material and Beam Technology (Fraunhofer-IWS), 01277 Dresden, Germany; bernd.schultrich@iws.fraunhofer.de

**Abstract:** This critical overview analyzes the relations between deposition conditions and structure for hydrogen-free carbon films, prepared by vacuum arc deposition. The manifold of film structures can be roughly divided into graphitic, nanostructured and amorphous films. Their detailed characterization uses advantageously $sp^3$ fraction, density, Raman peak ratio and the mechanical properties (Young's modulus and hardness). Vacuum arc deposition is based on energetic beams of carbon ions, where the film growth is mainly determined by ion energy and surface temperature. Both parameters can be clearly defined in the case of energy-selected carbon ion deposition, which thus represents a suitable reference method. In the case of vacuum arc deposition, the relation of the external controllable parameters (especially bias voltage and bulk temperature) with the internal growth conditions is more complex, e.g., due to the broad energy distribution, due to the varying "natural" ion energy and due to the surface heating by the ion bombardment. Nevertheless, some general trends of the structural development can be extracted. They are critically discussed and summarized in a hypothetical structural phase diagram in the energy-temperature plane.

**Keywords:** vacuum arc deposition; diamondlike carbon; $sp^3$ fraction; structural phase diagram





## 1. Introduction

Carbon atoms have the peculiarity to realize bonding states with different dimensionality with their four valence electrons: $sp^3$, $sp^2$ and $sp^1$ bonds. Tetrahedral $sp^3$ bonds lead to three-dimensional arrangements, trigonal $sp^2$ bonds to plane sheets and linear $sp^1$ bonds to chains. Physical vapor deposition (PVD) allows the preparation of pure carbon films (nominally) without any other elements as hydrogen and of structures with a broad combination of the various bonding types, especially in $sp^2$- and $sp^3$-bonded arrangements (Carbynelike $sp^1$ configurations are only realized by low-energy deposition, e.g., via gas-phase condensation, which is not considered in this overview). The resulting structures can roughly be divided into three groups (Table 1): graphitic' nanostructured; and amorphous films. Substantially, they correspond to the notations stage 1, stage 2 and stage 3, derived by Ferrari and Robertson from an analysis of the Raman spectra of vacuum arc deposited carbon films [1,2].

Graphitic carbon is built up of stacks of weakly bonded graphene layers, i.e., layers of $sp^2$-bonded hexagons (six-membered rings = aromatic rings). The ordered layers may have larger extensions as plane sheets (as in crystalline graphite) or as curved and distorted lamellae (as in turbostratic carbon) or may form arrangements of small graphitic clusters. The graphene layers may have a preferential orientation, especially parallel or perpendicular to the surface. The structural characterization demands information on the dimensions, the preferred orientations and the distortions of the graphene stacks and on the usually existing nanoporosity. Essential properties such as hardness and electrical resistivity are determined by the three-dimensional long-range connectivity of the $sp^2$ network.

Nanostructured carbon is mainly $sp^2$ bonded. It represents a composite of graphitic hexagons, of further $sp^2$ bonded atoms, disordered in olefinic chains and dimers, and of

some $sp^3$-bonded atoms. They may be characterized by the mean size $L_a$ of the graphitic clusters, the fraction $f_g$ of those $sp^2$-bonded atoms, which are arranged in hexagons and by the fraction s of $sp^3$-bonded atoms, typically below 20%.

Amorphous carbon consists mainly of a random network of $sp^3$- and $sp^2$-bonded atoms, where the three-dimensionally connected $sp^3$ bonds represent the strong skeleton. Only a few atoms are arranged in hexagons, typically below 10%. The $sp^3$ fraction represents the most important characteristic, which determines density ρ, Young's modulus E and hardness H. Approximately holds [3] (Chapter 6):

$$\rho(s) \approx (1.8 + 1.6 \times s) \text{ g/cm}^3 \tag{1}$$

$$E(s) \approx s \times 800 \text{ GPa} \tag{2}$$

$$E(\rho) \approx (\rho/(\text{g/cm}^3) - 1.8) \times 500 \text{ GPa} \tag{3}$$

$$H(s) \approx E/10 \approx s \times 80 \text{ GPa} \tag{4}$$

For tribological applications, amorphous carbon films with high $sp^3$ fractions $s \geq 50\%$ (ta-C, tetrahedrally bonded amorphous carbon) are of special interest. They are superhard with hardness values above 40 GPa.

**Table 1.** Types of carbon films.

| Type | Bonds | Structure |
|:---:|:---:|:---:|
| **graphitic** | $sp^2$ | extended stacks of graphene lamellae, possibly with preferential orientation, possibly with nano-/microporosity |
| **nanostructured** | $sp^2$ + <20% $sp^3$ | amorphous $sp^2$ + graphitic hexagons (+$sp^3$), possibly with nanoporosity |
| **amorphous** | $sp^3$ + $sp^2$ | amorphous $sp^3$ + $sp^2$ + (graphitic hexagons) |

The carbon structures result from the balance between the impact of the carbon ions and the subsequent structural relaxation. Hence they are mainly determined by the energy of the impinging particles and by the substrate temperature. The particle flux differs markedly for the various ion-beam and PVD methods (Table 2) [3] (Chapters 9, 14).

**Table 2.** Methods for the deposition of hydrogen-free carbon films (IBAD = ion beam assisted deposition: ion beam assisted evaporation, double ion beam sputtering, magnetron sputtering).

| Method | Deposition | Growth |
|:---:|:---:|:---:|
| evaporation, ion beam sputtering | C atoms | surface condensation |
| ion beam assisted deposition (IBAD) | C atoms + energetic noble gas ions | surface condensation + atomic peening |
| pulsed laser deposition (PLD) | $C_1 + C_2 + C_3$ atoms + $C_1^+ + C_2^+ + C_3^+$ ions | surface condensation + atomic peening + subplantation |
| vacuum arc ion beam deposition | $C^+$ (+ $C^{++}$) ions $C^+$ ions | subplantation subplantation |

The following discussion concentrates on the vacuum arc deposition: (1) It represents the most efficient method for the preparation of superhard ta-C films. (2) The vacuum arc works with a pure flux of carbon ions with a limited energy distribution, in contrast to ion beam-assisted deposition (IBAD) and pulsed laser deposition (PLD), which use a mixture of various atoms and ions with widely differing energies. Thus, the growth conditions

are less complex. (3) Nevertheless, a broad field of carbon structures can be realized by modified deposition conditions.

## 2. Characterization Methods

For the characterization of carbon films, various parameters and numerous experimental methods for their determination have been used [3–9]. Their efficient application in quality control and process optimization under industry-oriented conditions demands the applicability of real parts or at least of comparable reference substrates. For the often-used silicon substrates, e.g., for electron energy loss spectroscopy (EELS) for the determination of $sp^3$ fraction and density, the deviating heat dissipation must be considered. This can lead to markedly differing surface temperatures and correspondingly modified relaxation conditions. The following discussion concentrates on the most significant characteristics: density $\rho$, $sp^3$ fraction s, Raman peak ratio $I_D/I_G$, position of the Raman G peak $\nu_G$, Young's modulus E and hardness H.

### 2.1. Mass Density

The mass density is usually determined from the plasmon peak $\varepsilon_p \sim (n_e/m_e{}^*)^{1/2}$ near 30 eV in the EELS spectrum. Because each carbon atom contributes four valence electrons, the atomic density n is related to the density $n_e$ of the outer electrons by $n = n_e/4$. The effective electron mass $m_e{}^*$ can be estimated by fitting the crystalline carbon modifications with 100% $sp^3$ bonds (diamond) and 100% $sp^2$ bonds (graphite): $m_e{}^* \approx 0.853$ $m_e$ (by comparing densities determined by X-ray diffraction (XRD) and EELS for very different types of amorphous carbon, Ferrari and co-workers suggested a similar value $m^* = 0.87\ m_e$ [10]). The atomic density n and the mass density $\rho$ can now be derived from the plasmon energy $\varepsilon_p$ by

$$\rho \approx 0.00312\ \text{g/cm}^3\ (\varepsilon_p/\text{eV})^2 \tag{5}$$

Voids above a typical plasmon wavelength of 0.5 nm does not contribute to the plasmon signal [11]. Hence, the plasmon energy and the resulting local density reflect the atomic concentration in clusters of about ten carbon atoms. Carbon films with a plasmon energy above 27 eV, corresponding to a local density $\rho > \rho_g = 2.26\ \text{g/cm}^2$, have denser atomic environments than graphite. Hence, they must content a lot of $sp^3$-bonded atoms. The $sp^3$ fraction can be estimated from (1) with the local density, if the films are sufficiently amorphized by an intense ion impact. In this case, the density $\rho = 2.26\ \text{g/cm}^2$ corresponds to a $sp^3$ fraction of already $s \approx 29\%$, whereas the same density is also realized in perfect graphite without any $sp^3$ bonds. This hints at a loosening of the atomic arrangements in disordered $sp^2$-bonded structures. For local densities below 2.26 g/cm$^3$, densification may reflect increasing diamond-likeness (higher $sp^3$ fraction), increasing graphitization (higher graphitic order) or reduced porosity.

### 2.2. $sp^3$ Fraction

The standard method for the determination of the $sp^3$ fraction uses the high-energy peak around 290 eV of the EELS investigations [12–16]. For carbon films, the main features in the EELS C(1s) K-absorption edge come from the transition from the 1s core level to empty antibonding states: to $\pi^*$ at $E_{\pi*} \approx 284$ eV and to $\sigma^*$ at $E_{\sigma*} \approx 293$ eV. Usually, it is assumed that the strength of the $\pi^*$ peak is proportional to the number of $sp^2$-bonded atoms. The normalized intensity around the $\pi^*$ peak is integrated over a small energy window of about 5 eV between the onset of the peak up to the minimum between $\pi^*$ and $\sigma^*$ peak. This value $I_{aC}$ is divided by the corresponding integral $I_{sp2}$ for a reference sample containing only $sp^2$ bonds. The $sp^3$ fraction is then given by

$$s = 1 - f_{sp2} \approx 1 - I_{aC}/I_{sp2} \tag{6}$$

This procedure may overestimate the $sp^2$ bonds, because the $\pi^*$ integral may also contain some contributions from the $\sigma^*$ peak. A critical problem is the right specification of

the energy windows. For thicker films, the increasing contribution of multiple scattering must be considered. For a ta-C foil of circa 50 nm thickness, investigated by 200 keV electrons, it achieves about 8% of the total scattered intensity. The single scattering distribution must then be extracted from the raw data by a suitable deconvolution technique [14]. An additional uncertainty is introduced by the used collection angle. A downshift of about 1 eV has been reported for the transition from conventional collection angle (10 mrad) to a reduced angle with higher resolution [17]. Hence, the necessary preparation of sufficient thin foils (by dissolving the silicon substrate), the used measuring technique and deviating evaluation procedures induce some uncertainty of the extracted $sp^3$ values.

The laborious sample preparation can be avoided by using surface analytical methods, such as Auger electron spectroscopy (AES) or X-ray photoelectron spectroscopy (XPS) [18,19]. However, the limited sampling depth must be considered. The released and recorded electrons have for carbon films only a very small escape depth, leading to an information depth of about 3 nm for AES and 10 nm for XPS [20].

### 2.3. Raman Spectroscopy

For the usual excitation with visible light, the Raman spectrum of PVD carbon films is (due to resonance enhancement) completely determined by the $sp^2$ bonded carbon atoms, notwithstanding a fraction of up to 90% $sp^3$ bonds in disordered arrangements. (The diamond peak at 1322 cm$^{-1}$ occurs only for $sp^3$ bonds in crystalline diamond grains). Their stretching oscillations induce the G peak at 1584 cm$^{-1}$ in graphite [21]. The radial breathing of the graphitic hexagons (=aromatic rings) is only Raman active, if their symmetry is disturbed, leading to the D peak at about 1360 cm$^{-1}$. The main characteristics of the Raman spectrum are the positions $\nu_G$ and $\nu_D$ of the G peak and the D peak, their widths, expressed by the full width half maximum (FWHM) $\Delta\nu_G$ and $\Delta\nu_D$, and the integrated intensity ratio $I_D/I_G$ of both peaks.

The width of both peaks increases with the stronger distortion of the local structures. However, the peaks are also broadened by local structural variations due to intentional or unintentional modifications of the deposition conditions. Typically, the D peak is twice as broad as the G peak. In comparison to the G peak, the D peak position is less clearly defined and shows lesser correlations with the deposition conditions. Hence, the Raman ratio $I_D/I_G$ and the G peak position represent the most significant Raman characteristics.

In vacuum arc-deposited films, the G peak width varies between 100 and 200 cm$^{-1}$ and both peaks (separated by less than 200 cm$^{-1}$) merge usually to a common asymmetric curve. The D peak emerges only as an asymmetry or as a light shoulder (only at a high temperature deposition above 300 °C did the two separated peaks appear). The separation into the contributions of the both peaks depends on the used fit procedure. Usually, the G and the D peak are fitted by two symmetric functions, two Gauss distributions or a Gaussian (for the D peak) and a Lorentzian (for the G Peak). For comparability, only such evaluations are considered in the following. Sometimes, the G peak (or the complete spectrum) has been fitted by an asymmetric Breit-Wigner-Fano distribution (BWF). Due to the asymmetry of the BWF distributions with stronger tails on the low-frequency side, some signals, which would belong to the D peak for symmetric fit functions, are now assigned to the G peak [1,22]. Hence, both peak positions are sometimes shifted towards lower frequencies and the Raman ratio is sometimes reduced.

Some investigations observed a shift of the G-peak position towards higher wave numbers only by compressive stresses σ [23,24]:

$$\nu_G(\sigma) \approx \nu_G(\sigma = 0) + \alpha \times \sigma \tag{7}$$

Systematic investigations of the stress release over a broad stress range lead to $\alpha$ = 4.1 cm$^{-1}$/GPa, which is near to the value for graphitic materials with randomly oriented planes under biaxial stress (4.9 cm$^{-1}$/GPa) [24]. For compressive stresses around 10 GPa in highly $sp^3$-bonded ta-C films, this means an upwards shifts of 40 cm$^{-1}$. Correspondingly, the influence of structural modifications on the G peak position could be completely

overlaid by parallel variations of the compressive stresses. However, other investigations showed negligible effects of the stress relaxation on the Raman spectrum: Thin ta-C films with a high $sp^3$ fraction of 87% have been prepared by a filtered vacuum arc deposition and were subsequently annealed in a vacuum [25]. For annealing temperatures of about 600 °C, the compressive stresses fall from 11 GPa to nearly zero, whereas the Raman spectrum remains nearly unchanged.

Instead of the ratio $I_D/I_G$ of the peak areas, sometimes, the ratio of the peak heights $H_D/H_G$ is used (e.g., in the early works [26,27] and for Raman fits with the BWF distribution). Both quantities are approximately related by

$$I_D/I_G \approx (\Delta\nu_D \times H_D)/(\Delta\nu_G \times H_G) \approx 2\, H_D/H_G$$

Graphitic materials consist of stacks of aromatic rings. The peak intensity ratio $I_D/I_G$ rises with the increasing number of distorted rings. Due to the weak interlayer interaction, it is sufficient to consider the structure within the layers. The ring systems have finite dimensions, characterized by the lateral size $L_a$. For such a ring cluster, the G peak is caused by all of the enclosed atoms. Their number is proportional to the area of the ring cluster: $I_G \sim L_a^2$. The distorted rings are mainly positioned along the rim of the ring cluster. The number of these rings, contributing to the D peak, is proportional to the circumference: $I_D \sim L_a$. Consequently, the peak ratio is inversely proportional to the cluster dimensions:

$$I_D/I_G(\xi) \approx c_{DG}/L_a(\xi) \quad \text{(for graphitic materials)} \tag{8}$$

where $\xi$ characterizes the increasing disorder. This is just the relation, experimentally found out by Tuinstra and Koenig, by comparing the results of Raman spectroscopy and X-ray diffraction [26,27]:

$$H_D/H_G(\xi) \approx c_{DG}'/L_a(\xi) \approx 4.4\ \text{nm}/L_a \qquad \text{(for } \lambda = 532\ \text{nm)} \tag{9}$$

with $I_D/I_G \approx 2\, H_D/H_G$. For deviating wavelength $\lambda$ of the Raman excitation, the coefficient increases according to $c_{DG}' \sim \lambda^2$ for carbon materials with small $L_a$ near 2 nm [28]. Investigations of various types of carbon materials support the general trend of rising disorder for the increasing Raman ratio, but revealed a broad scatter of the coefficients $c_{DG}$ between 4 and 15 [29] and of $c_{DG}'$ between 3 and 8 [28] (corresponding to $c_{DG} \approx 6$–16).

The simplified picture of distorted rings along the periphery and non-distorted rings in the interior holds only for larger clusters. For very small clusters $L_a < L_a^*$, all rings have a non-ideal environment and contribute correspondingly to the Raman breathing mode. Hence, the D peak should not further increase for such small clusters and the Raman ratio should tend to a constant value $I_D/I_G \approx c_{DG}/L_a^*$. Unfortunately, the $L_a$-range below about 2 nm is experimentally not accessible.

In nanostructured films, only a fraction $f_g$ of the $sp^2$-bonded atoms are included in graphitic rings. An increasing number of the atoms are arranged in short chains, dimers or amorphous structures, leading to

$$I_D/I_G(\xi) \approx f_g(\xi)\, c_{DG}/L_a(\xi) \qquad \text{(for nanostructured films)} \tag{10}$$

Both parameters $L_a(\xi)$ and $f_g(\xi)$ in Equation (10) decrease with the rising disorder $\xi$, resulting in a maximum at $\xi_m$. For lower disorder $\xi < \xi_m$, the intensity ratio decreases due to the less number of distorted rings. For higher disorder $\xi > \xi_m$, the intensity ratio decreases due to the reduction of $sp^2$ atoms, bonded in hexagons. The dimensions of the ring clusters are very small, below $L_a^*$. The comparison of Equations (8) and (10) yields a maximum of the Raman ratio $\max(I_D/I_G) \approx f_g(\xi_m)\, c_{DG}/L_a(\xi_m)$ at the transition from the graphitic to the nanostructured films. Maximum Raman ratios up to about 6 have been observed [30–32]. Tentatively $\xi_m \approx \xi^*$ can be assumed.

*2.4. Young's Modulus/Hardness*

The reversible mechanical behavior is described by the Young's modulus E, and the irreversible deformation by the hardness H. In covalently bonded materials, both properties are strongly correlated. For carbon films holds approximately

$$H \approx E/10 \tag{11}$$

In contrast to the $sp^3$ fraction and the Raman characteristics, which only reflect the local environment, Young's modulus and hardness are determined by the global connectivity of the bond network. Usually, they are derived in instrumented nanoindentation tests from the relation between penetration depth vs. force (see e.g., [33–36]). To exclude interference with the underlying substrate, the penetration depth must be sufficiently small, typically below one tenth of the film thickness. The indentation test allows local investigations, but demands sufficient smooth surfaces. The main problem consists of the influence of the tip geometry, which must be thoroughly corrected.

For the Young's modulus, an alternative approach is given by the evaluation of surface acoustic waves (SAW) [37–39]. The penetration depth of the ultrasonic waves in the film-substrate composite scales with their wavelength $\lambda$. Thus, the respective influence of film and substrate on the phase velocity c varies with the frequency $f = c/\lambda$. By fitting the dispersion curve c(f), the Young's modulus of the carbon film can be derived, in the case of sufficient non-linearity and also film density and thickness. The method works also on rough surfaces. Indentation and SAW yield usually comparable results. Deviations occur for anisotropic films, because indentation reflects mainly the stiffness in the normal direction and the surface waves mainly in the lateral direction [40].

Table 3 summarizes the methods, recommended for the efficient characterization of hydrogen-free carbon films.

**Table 3.** Advantageous characterization methods for hydrogen-free carbon films.

| Characteristic | Method | Applicability |
|---|---|---|
| $sp^3$ fraction s | EELS (high energy) | thin films, detached from the Si substrate |
| local density $\rho$ | EELS (plasmon peak) | thin films, detached from the Si substrate |
| global density $\rho$ | surface waves | real substrates, also thick films, also rough films |
| Raman peak ratio $I_D/I_G$ | Raman spectroscopy | also rough films |
| G peak position $\nu_G$ | | |
| Young's modulus E | surface waves | also rough films |
| Young's modulus E | indentation | smooth films |
| Hardness H | indentation | smooth films |

## 3. The Reference Case: Ion Beam Deposition

The vacuum arc deposition (VAD) works with a fully ionized carbon plasma, consisting nearly completely of $C^+$ ions. The resulting structure is determined by the energy and the angle of incidence of the ion beam and by the surface temperature. In the mass selected ion beam deposition (MSIBD), these parameters can be adjusted in a very controlled manner, in contrast to the more complex conditions in VAD. Thus, MSIBD represents a suitable reference case.

In MSIBD, the carbon ions are generated within a separated discharge chamber, e.g., from $CO_2$ or CO. They are extracted from the ionization region and simultaneously accelerated by a high voltage of some kV or even some ten kV. The interesting ion species are selected by an electromagnetic mass analyzer. Finally, the ion beam is decelerated by a suitable countervoltage to the desired ion energy. The reduction of the ion energy is impeded by the increasing space-charge spreading for lower energies. It demands very low ion current densities, resulting in correspondingly tiny deposition rates. For carbon ions below 100 eV, maximum ion arrival rates of 100 $\mu A/cm^2$ (corresponding to deposition rates of about 0.06 nm/s) have been estimated [41]. A differential pumping system allows

very low pressures within the deposition chamber, notwithstanding the necessary working pressure within the ion source. The co-deposition of any other species is widely excluded by the electromagnetic mass analyzer and by the high vacuum or (even better) ultrahigh vacuum within the deposition chamber.

### 3.1. Ion Energy

For ion energies $\varepsilon$ up to the optimum energy of about 100 eV for maximum $sp^3$ fraction, the results of the different MSIBD groups yield a coincident picture (Figure 1a) [42–44]: At an ion energy of 10 eV, corresponding to the threshold value for the subplantation process, the films have a $sp^3$ content of about 10%. The a-C/ta-C boundary (for a $sp^3$ fraction s = 0.5) is achieved for an ion energy of 24 eV. Above 30 eV, the slope is weakened and the $sp^3$ fraction achieves a plateau above 80%. The energy range up to 100 eV can be described by an exponential

$$s(\varepsilon) \approx s_{thr} + (s_{max} - s_{thr})\{1 - \exp(-(\varepsilon - \varepsilon_{thr})/\varepsilon_0)\} \qquad \text{for } 10 \text{ eV} \leq \varepsilon \leq 100 \text{ eV}$$
$$\approx 0.1 + 0.75\{1 - \exp(-(\varepsilon/\text{eV} - 10)/18)\} \tag{12}$$

with the threshold values $\varepsilon_{thr} = 10$ eV and $s_{thr} = s(\varepsilon_{thr}) = 0.1$, the maximum $sp^3$ fraction $s_{max} \approx 0.85$ and $\varepsilon_0 = 18$ eV. Between 10 and 30 eV, the $sp^3$ fraction increases linearly with the ion energy. Approximately holds

$$s \approx 0.028 \, \varepsilon \, /\text{eV} - 0.18 \qquad \text{for } 10 \text{ eV} \leq \varepsilon \leq 30 \text{ eV} \tag{13}$$

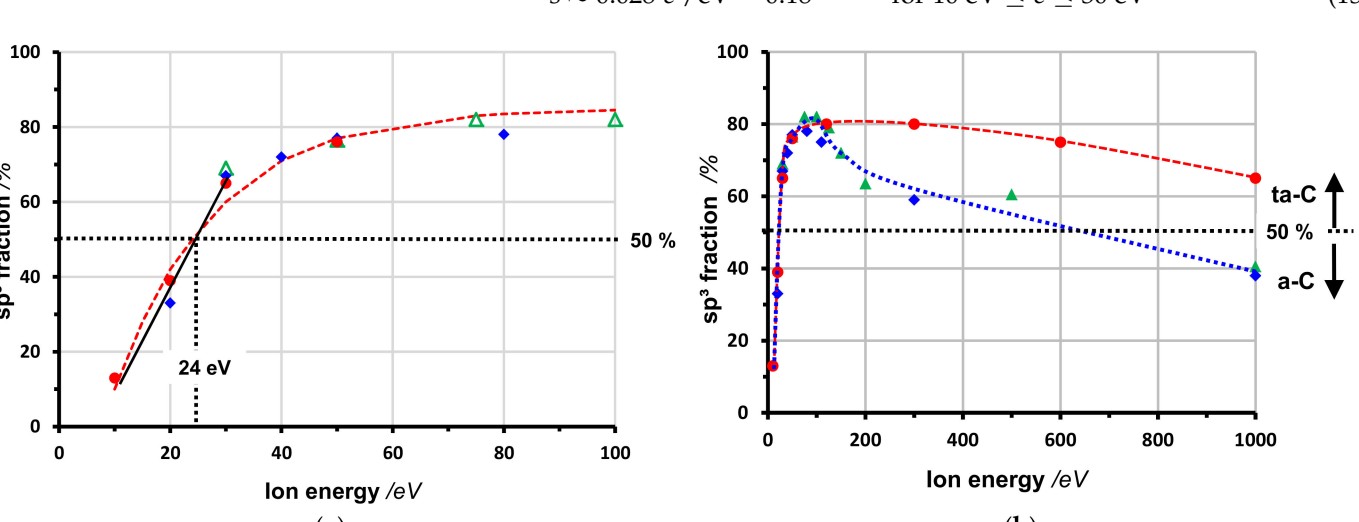

(a)          (b)

**Figure 1.** $sp^3$ fraction of carbon films, prepared by MSIBD at room temperature, in dependence on the ion energy (data from [42] (blue diamonds), [43] (red circles), [44] (green triangles)). (**a**) Up to medium ion energies. The dotted red line and the solid black line describe the exponential approximation Equation (12) and the linear slope Equation (13), respectively. (**b**) Extended energy range.

The steep increase of the $sp^3$ fraction for ion energies between 10 and 40 eV is caused by the change in the growth process, directly observable by the deviating surface topography: At low energies (typically below 10 eV), the impinging carbon ions remain on the surface. Due to their high surface mobility, they form in the beginning separated islands, which then coalescence to a continuous, but a rough film. At higher ion energies, the impinging ions penetrate the surface (subplantation) and the film grows from the interior in a very smooth manner. Increasing the energy from 5 to 50 eV, the mean roughness falls from 3.5 nm to only 0.2 nm and remains on this level up to 10 keV (Figure 2) [45].

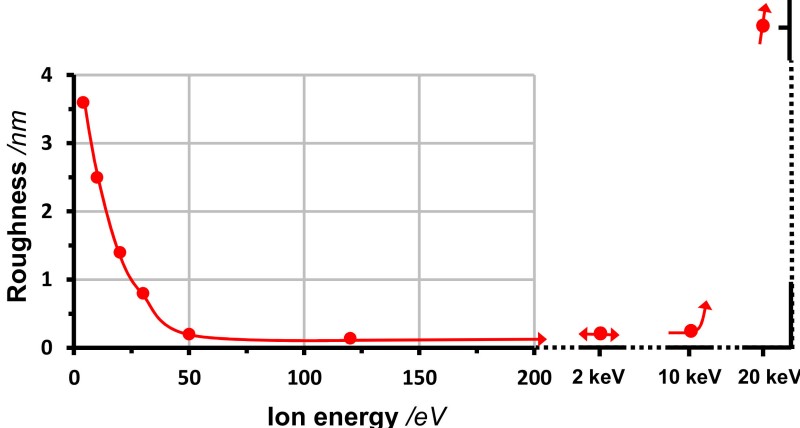

**Figure 2.** Roughness of carbon films, deposited by MSIBD at room temperature, in dependence on the ion energy. Film thickness about 100 nm (data from [45]).

For ion energies above 300 eV, the radiation-enhanced diffusion promotes increasingly the relaxation of the strongly stressed structure. The $sp^3$ fraction falls gradually (Figure 1b). Even in the kiloelectronvolt range, it remains high with s(2 keV) = 52% and s(10 keV) = 44% [45]. However, for 20 keV, the $sp^3$ bonds represent only a minority of about 9%. The corresponding alteration of the growth process is indicated by the rise of the roughness from 0.3 nm at 10 keV to 30 nm at 20 keV (Figure 2). Up to 10 keV no graphitic ordering has been observed apart from the $sp^2$-bonded top layer of about 1 nm thickness [46]. With 15 keV ions, graphene layers show a preferred orientation perpendicular to the substrate surface. At 20 keV deposition, a layered structure occurs with the perpendicular orientation of the bottom layer and the parallel orientation of the top layer.

Sometimes, a steep decline of the $sp^3$ fraction to about 60% has been observed already at much lower ion energies of 200–500 eV (Figure 1b) [42,44,47]. This can be attributed to the damaging effect of non-eliminated keV neutrals and the resulting irradiation-induced relaxation.

Generally, $sp^3$ fraction, density, Young's modulus and hardness show a similar dependence on the ion energy (Figure 3).

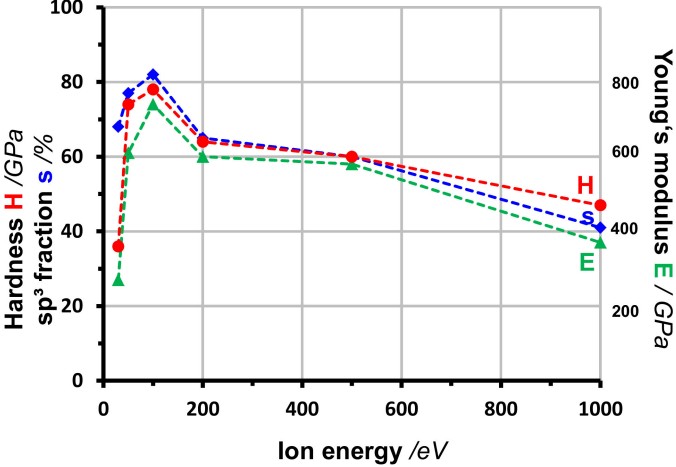

**Figure 3.** $sp^3$ fraction, Young's modulus and hardness of carbon films, deposited by MSIBD at room temperature, in dependence on the ion energy (data from [47]).

The development of the Raman spectrum reflects these structural modifications (Figure 4) [22]: The peak asymmetry at the 10 eV deposition corresponds to the essential contribution of the D peak (nanostructured carbon). It diminishes higher energies, leading to a completely symmetric peak, i.e., a vanishing D peak, between 120 and 300 eV

(amorphous carbon). For higher energies between 800 and 10 keV, the asymmetry occurs again increasingly (nanostructured carbon). At 20 keV, finally, separated D and G peaks are clearly identifiably. The correspondingly small peak width hints to rather well ordered structures (graphitic carbon). Additional information is given by the peak around 950 cm$^{-1}$, representing the second order peak of the underlying silicon substrate. Its height shows the good transparency of the 100 nm films for deposition with 120–300 eV ions and the increasing absorption beyond this optimum range.

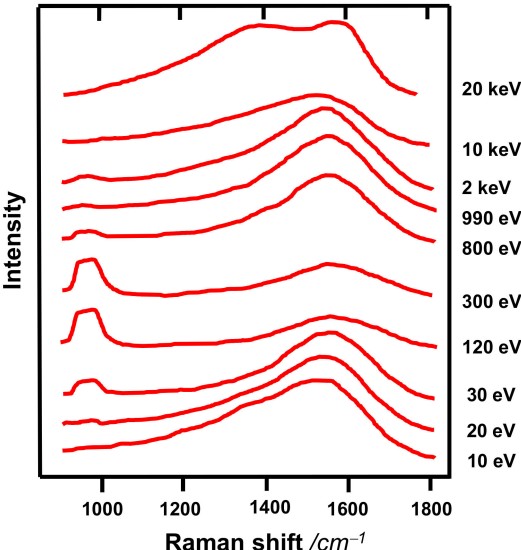

**Figure 4.** Raman spectrum of carbon films prepared by MSIBD at room temperature with various ion energies (data from [22]).

*3.2. Deposition Temperature*

At higher temperatures, the sp$^3$ fraction decreases above a critical temperature $T_c$. For optimum ion energies around 100 eV, the transition ta-C to a-C occurs at about 150 °C. The energetic ions penetrate beneath the surface (subplantation), generating sp$^3$ bonds. However, driven by the very high compressive stresses within the film, these interstitials migrate to the surface, where they condense as in the usual film growth with low-energy atoms. Without the high-pressure conditions in the interior, they develop sp$^2$ bonds. According to the changed growth mechanism, the roughness grows rapidly beyond the critical temperature (Figure 5) [48]. For higher ion energies, the ions penetrate deeper and $T_c$ shifts to some higher values, e.g., from 150 °C for 120 eV ions to above 165 °C for energies of 300 eV [14].

Deposition at much higher temperatures of 860 °C results for 100 eV ions in very coarse films: They consist of particles of about 500 nm for a mean thickness of 300 nm [49]. In contrast, a 500 eV deposition yielded rather smooth films at 860 °C. The bombardment with the 500 eV ions seems to suppress the coalescence of the nuclei to larger entities. In both cases, the electron diffraction revealed a graphitic structure with grains of about 4 nm, in comparison to 1–2 nm at 330 °C deposition.

The graphitic nature of the high-temperature films is clearly demonstrated by the Raman spectra: In contrast to the nearly symmetric peak for the room-temperature deposition (corresponding to amorphous carbon), the spectra at 400 and 800 °C show two clearly separated peaks, the D peak at 1345 cm$^{-1}$ and the G peak at 1595 cm$^{-1}$ (Figure 6) [41]. The decreasing peak width hints at the increasing ordering in the graphitic films. The high-temperature Raman spectra are nearly independent of the used ion energies between 15 and 500 eV. This means that the carbon structure in high-temperature deposition in this range is mainly determined by the temperature-induced relaxation processes.

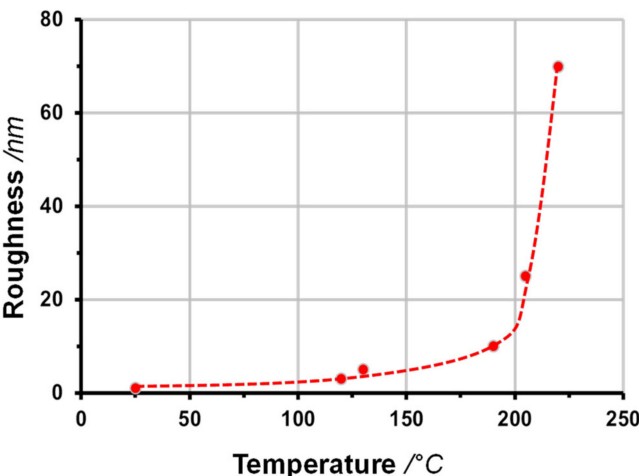

**Figure 5.** Roughness of carbon films, deposited by MSIBD with 120 eV carbon ions, in dependence on the deposition temperature. Film thickness about 100 nm. Data from [48].

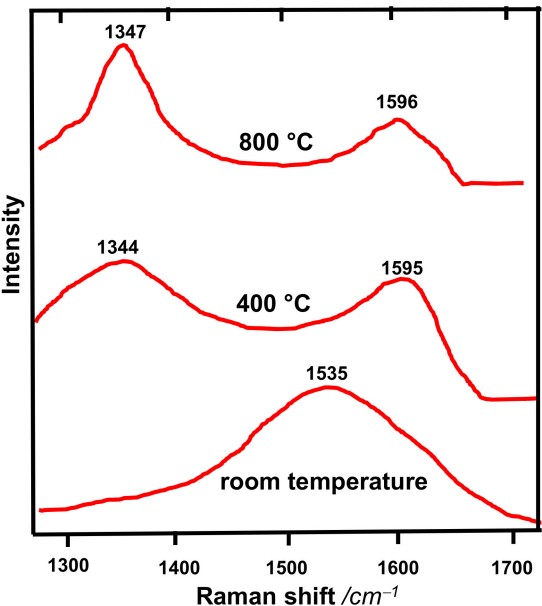

**Figure 6.** Raman spectrum of carbon films deposited by MSIBD at various temperatures (data from [41]).

High resolution transmission electron microscopy (HRTEM) reveals the high degree of graphitic ordering, which is already realized at temperatures between 200 and 300 °C [50–52]. Due to the high compressive stresses by the ion impact, the graphene layers are preferentially oriented perpendicular to the surface. The curved stacks resemble ensembles of multiwalled nanotubes.

For deposition with carbon ions of about 100 eV at enhanced temperatures, density and Young's modulus remain initially unchanged and start to fall at about 150 °C (Figure 7a). At about 170 °C, they are reduced to 2.6 g/cm$^3$ and 400 GPa, respectively. According to Equations (1) and (2), these values correspond to a sp$^3$ fraction s of 50%, i.e., the transition from ta-C to a-C films. Figure 7b shows that the combination E($\rho$) of both relations E(T) and $\rho$(T) in Equation (3) describes well the experimental data.

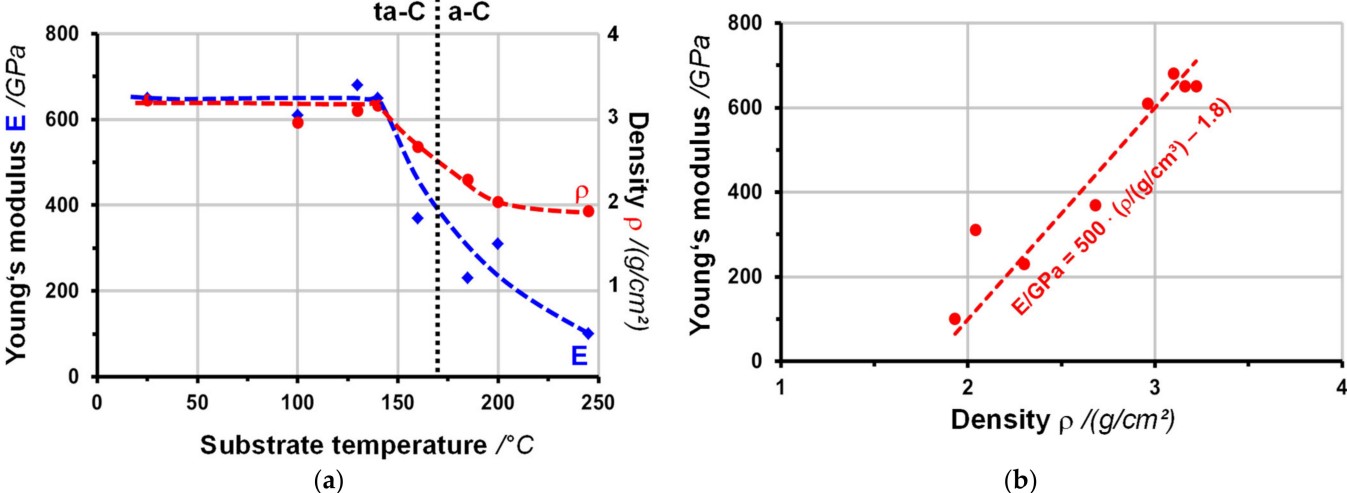

**Figure 7.** (**a**) Young's modulus and density of carbon films, deposited by MSIBD with 120 eV ions, in dependence on the substrate temperature (data from [52,53]). E and ρ have been determined by laser acoustics. (**b**) Relation between Young's modulus and density of carbon films, deposited by MSIBD with 120 eV ions at temperatures between room temperature and 250 °C (data from (**a**)). Included is the relation Equation (3) for E(ρ).

## 4. Vacuum Arc Deposition

### 4.1. Deposition Conditions

The plasma of vacuum arc discharges with graphite cathodes and is completely ionized. Usually, the expanding ion beam consists of $C^+$ ions and constitutes about one tenth of the arc current for the mostly used arc currents below 1000 A. For higher arc currents above 1 kA, as they are realized in pulsed discharges, the ratio is even higher, up to 20% and an increasing fraction of $C^{++}$ ions occurs. With typical mean ion currents around 10 A, the vacuum arc represents a very powerful ion source.

A principal problem of the vacuum arc discharges with graphite cathodes consists in the local stationarity of the cathode spots, the local centres of the carbon emission. Apart from the irregular erosion of the cathode surface (with correspondingly low target utilization), it leads to the emission of target fragments of nanometer or even micrometer dimensions (macroparticles), which generate growth defects in the film. To minimize these detrimental inclusions, various concepts, partially in combination, are used [3] (Chapter 10):

-  dc arcs (arc current typically around 100 A) with strong magnetic fields and optimized cathode geometry;
-  pulsed arcs with high arc currents in the kiloampere range;
-  pulsed arcs with controllably displaced spot ignition (e.g., Laser-Arc);
-  separation of the macroparticles by curved magnetic fields (filtered arc).

#### 4.1.1. Ion Energy

For usual arc currents below 1 kA, the carbon ions propagate in the plasma with a mean velocity of about 18,000 m/s, corresponding to a "natural" kinetic energy $\varepsilon_0 \approx 20$ eV within the arc plasma [54,55]. For currents in the kiloampere range or for strong magnetic fields at the cathode, the mean ion energy scales roughly with the increasing arc voltage $U_{arc}$:

$$\varepsilon_0 \approx e\, U_{arc} \tag{14}$$

For the energy of the ions, impinging the substrate, the potential difference between plasma and substrate must be considered:

$$\varepsilon_i = \varepsilon_0 + e\,(\varphi_p - \varphi_{sub}) \tag{15}$$

Here, $\varphi_p$ and $\varphi_{sub}$ denote the respective potentials against ground, which is usually directly connected with the anode and the chamber walls. Mostly, the plasma potential is only some volts above the anode potential. A sufficient conducting substrate is usually grounded or it is stressed by a negative bias voltage $\varphi_{sub} = -U_{bias}$ against the ground. In the case of insulating or insulated substrates (or substrate holders), the substrate takes the floating potential: $\varphi_{sub} = \varphi_{fl}$. Generally, the floating potential is by 10–20 V below the plasma potential. For filtered carbon arcs, values between −8 and −23 V have been determined, depending on the magnetic field, the duct bias and the arc current [56]. Thus, the ions, impacting floating substrates, get an extra energy in comparison to grounded substrates. For instance, carbon films, deposited with 1 ms pulses of some hundred amperes on floating substrates showed a $sp^3$ fraction of 64% in comparison to 47% on grounded substrates [57]. According to Equation (12), they correspond to ion energies of 33 and 22 eV, respectively.

For the elimination of the graphitic macroparticles, the carbon plasma is often guided through a magnetic filter unit with axial magnetic fields of some ten mT and a positive duct bias of about 15–20 V. For a broad variation of field strength and duct bias, variations of the ion energies between 24 and 33 eV have been observed in the plasma near the exit of a typical 90° filter [56]. The effect has been explained by the acceleration (and vice versa deceleration) of the ions, when they run into regions with weaker (or stronger) magnetic fields [54]. In this way, mean ion energies $\varepsilon$ between 10 and 120 eV have been realized within the carbon plasma with special magnetic field configurations.

Under usual conditions, mean ion energies around 20 eV and corresponding $sp^3$ fractions of 40–50% can be expected for substrates on anode potential. However, the influence of the plasma potential, temporal variations (in pulsed discharges), magnetic field effects (for filtered or steered arcs) and insufficient charge dissipation (e.g., floating substrates) may lead to deviating ion energies. According to Equation (12), even small changes by $\pm 5$ eV result in essentially altered $sp^3$ fractions of s(15 eV) = 28% and s(25 eV) = 52%, respectively. The sensitivity of the carbon structures in this energy range is clearly demonstrated in [58]: With filtered arc deposition on non-biased substrates, $sp^3$ fractions (from EELS) of 37% and 52% have been realized for films, prepared under (nominally) identical conditions in different runs. The measured plasmon energies of 27.0 and 29.8 eV correspond (according to Equation (5)) to densities of 2.3 and 2.8 g/cm$^3$, confirming the essential structural variation.

In pulsed discharges, the temporal variations and the now possible high currents up to the kiloampere range must be considered: the early stage of the arc discharge is characterized by the arc-typical high currents, but is combined with high voltages up to the kilovolt range ("spark phase"). Due to the high power input, the plasma is excessively heated. The ions get very high energies and higher ion charges $C^{n+}$ occur. The initial excitation fades exponentially with a time constant of 30–70 µs [59]. For a 300 A carbon arc, 8.6% $C^{++}$ has been observed after 3 µs in contrast to 100% $C^+$ after 250 µs. In short-pulsed high current arc discharges with 7.5–10 kA in 30 µs, even a majority of higher ion charges has been found: 73% $C^{3+}$ with energies of 250 eV, 23% $C^{++}$ with 110 eV and only 4% $C^+$ with 32 eV [60]. The transition to a stationary plasma within the cathode–anode interspace is strongly influenced by the electrode configuration. The weight of the transient phase increases for shorter pulse length and for longer transit times, i.e., larger electrode distances. With 15 µs pulses of 10 kA, the mean energy of the carbon ions could be raised from 23 up to 750 eV, only by extending the distance from below 2.5 to 4.6 cm [61].

For Laser-Arc deposition with pulse currents up to 1 kA on grounded substrates, mean ion energies around 20 eV have been determined for a grounded anode ("anodic Laser-Arc", Figure 8). For higher arc currents, the ion energy increases slightly in accordance with the rising arc voltage. During the pulse, the arc voltage breaks down from the high open-circuit voltage (typically between 100 and 400 V) down to some ten volts. For example, for the sinusoidal arc currents of 100 µs in [62], the arc voltage amounts (up to 1 kA peak currents) to 70–80 V for the first 50 µs and falls then to the typical 20 V.

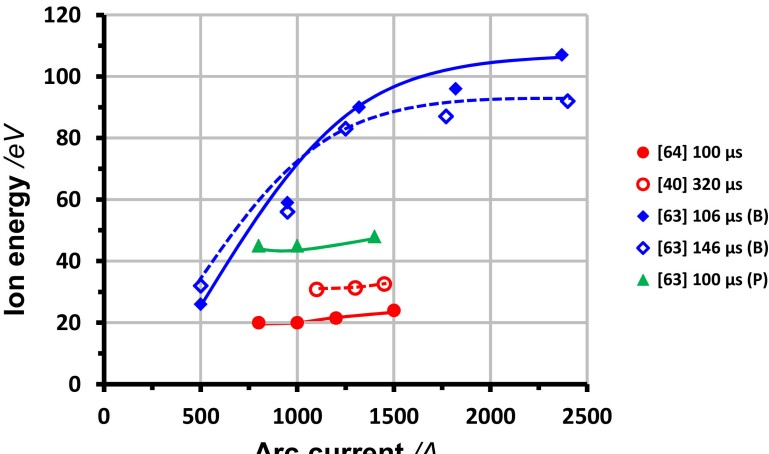

**Figure 8.** Energy of the carbon ions, measured against ground, vs. arc current maximum for Laser-Arc depositions. Cathodic Laser-Arc [63] with different coating units (B, P), anodic Laser-Arc [64] and filtered anodic Laser-Arc [40].

Generally, the influence of the initial high-voltage stage increases with the ratio of the time constant for the voltage drop (depending on the power supply and the anode-cathode configuration) to the transit time of the discharge. Hence, the anode geometry has in Laser-Arc devices, where the cathode-anode distance is only in the range of few millimeters, an essential influence.

If the cathode (instead of the anode) is grounded ("cathodic Laser-Arc"), the ion energy on a grounded substrate increases under else identical conditions (Figure 8), because the substrate is now on the cathode potential and thus negative in comparison to the anode. The difference is given by the arc voltage, which works (for a grounded substrate) as (a temporarily variable) substrate bias. Equation (14) yields approximately $\varepsilon \approx 2$ e $U_{arc}$.

In contrast to the well-defined ion beam in MSIBD with a small energy spread between 1 and 5 eV, the ion energy in the vacuum arc discharges has a broader distribution (Figure 9). For asymmetric distribution, the deviation of the mean energy from the peak energy must be considered. For instance, the curve from [40] in Figure 9b corresponds to a peak value of 20 eV and an essentially higher mean energy of 33 eV, whereas the curve from [63] (2370 A) yields a peak value of 100 eV and a lower mean energy of about 86 eV. For dc arcs without magnetic fields, a FWHM of about 18 eV has been measured [65,66]. By magnetic filtering, the energy distribution is often narrowed, because ions from a certain energy range are preferentially guided through the filter unit. In pulsed arc discharges, the temporal alterations leads to a broadening of the averaged distributions by the superposition of several stages. In the examples of Figure 9a, a FWHM of 7 eV has been determined for the filtered arc [67] and of 25–30 eV for a pulsed arc [64]. Other investigations of the carbon plasma in magnetically filtered arc discharges yield FWHM between 11 and 24 eV, depending on the magnetic field strength and the duct bias and the arc current [56].

Comparing the energy variations of different investigations, some uncertainties must be considered:

(1) The energy of the ions, impinging the substrate, is itself often not measured. Only the energy variation is controlled by the altered bias voltage. The additional energy on unbiased substrates is estimated, usually by about 20 eV, but may markedly deviates from this standard value.

(2) Different widths of the energy distribution, especially the high-energy tail, may influence the film growth.

(3) In the case of pulsed discharges and/or pulsed biasing, the instantaneous conditions vary during each pulse. They are only fragmentarily reflected by the mean ion energy.

(4) Effects from an unintended substrate heating by the energetic ion flux may overlay the direct influence of higher ion energies.

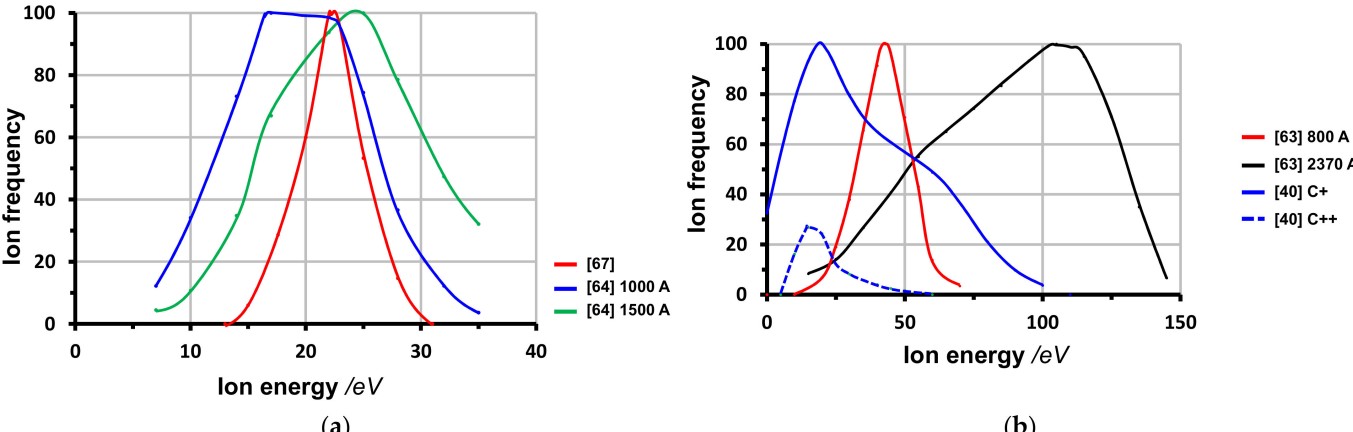

**Figure 9.** Energy distribution of the carbon ions, measured against ground (**a**) Filtered vacuum arc [67] and anodic Laser-Arc (pulse length 100 μs, pulse maximum 1 kA/1.5 kA [64]). (**b**) Cathodic Laser-Arc (pulse length 100 μs, pulse maximum 800 A/2370 A [63]) and filtered anodic Laser-Arc (pulse length 320 μs, pulse maximum 1450 A [40]).

### 4.1.2. Substrate Temperature

At low deposition rates, as in MSIBD, the temperature of the surface region can be very well controlled by a thermostating of the underlying substrate. However, high deposition rates, as they are realized in vacuum arcs, in combination with high ion energies, can lead to increasing temperatures during the deposition. The real deposition temperatures near the surface are then above the measured bulk temperatures. The deposition-induced surface heating is of particular importance for industrial coatings, where high deposition rates are usually an economic precondition.

### 4.1.3. Further Factors

The following discussion concentrates on the ion energy and the deposition temperature as the decisive factors for the formation of the carbon film structure. Moreover, there are some additional factors, which may influence the film growth:

- Inclined incidence: During the penetration of the energetic ions into the subsurface layer, they form transient $sp^3$ bonds due to the local high-pressure conditions. However, the pressure gradient drives the carbon atoms towards the stress-free surface, so that they partially relax their bond state to $sp^2$ [68]. Under inclined incidence with the angle $\vartheta$ against the surface normal, the penetration depth, i.e., the distance to the surface, is reduced by a factor $\cos\vartheta$, thus shortening the diffusion path. Additionally, the superficial layer is damaged by the oblique ion tracks, stimulating the carbon mobility by means of the irradiation-induced diffusion. Inclined incidence occurs inevitably at the deposition of complex shaped parts and at the deposition with rotating substrate holders. In the latter case, the films represent a multilayer stack according to the periodically changing angle.
- Deposition rate: Instantaneous very high deposition rates, as they are possible with pulsed high current arcs, limit the interval for structural relaxation. If they are combined with sufficient low repetition frequencies, additional heating by the intense ion impact can be minimized. For instance, for momentary rates of 1000 nm/s, the onset of the temperature-induced relaxation has been shifted from <200 °C up to 400 °C [69].
- Hydrogen content: Hydrogen may be picked up from the residual gas, especially at only moderate base pressures and with larger carbon deposits in the deposition chamber. However, in the case of vacuum arc deposition, most of the entered hydrogen is again released by the intense ion bombardment. Usually the unintentionally incorporated hydrogen remains below 1–2% with a negligible influence on structure and properties.

- Macroparticles: The graphitic macroparticles may not be completely eliminated from the plasma beam. Most of them are reflected, but some will be incorporated into the growing film. There they induce weakly embedded growth defects with a much larger volume. They may influence the film properties, especially the Raman spectra, the film roughness and the tribological behavior.

### 4.2. Ion Energy

In the case of MSIBD with well-defined ion energies, very low deposition rates and surface temperatures near room temperature, the dependence of the $sp^3$ fraction on the ion energy can be described by Equation (12): a steep (initially linear) rise above the threshold energy of about 10 eV, achieving ca. 75% $sp^3$ at about 50 eV and approaching the limiting value some above 80% at energies near 100 eV (Figure 1a).

For filtered vacuum arc deposition with ion energies between 50 and 120 eV, likewise only a small rise of the $sp^3$ fraction with the ion energy is observed (Figure 10a). The spread of the maximum values between 75% and 85% may be caused by deviating evaluation procedures in the EELS determination or by the influence of different energy distributions (especially the high-energy tails). For low bias voltage or even unbiased substrates, the values lie partially along the MSIBD curve, but partially they are much higher, at around 80% without bias. This could arise from the high-energy ions in extended ion distributions. If the ion energy has been only estimated from the reported bias voltage, the difference could also be caused by higher unbiased ion energies (in comparison to the usual 20 eV) due to an unconsidered acceleration in the inhomogeneous magnetic field within the filter unit.

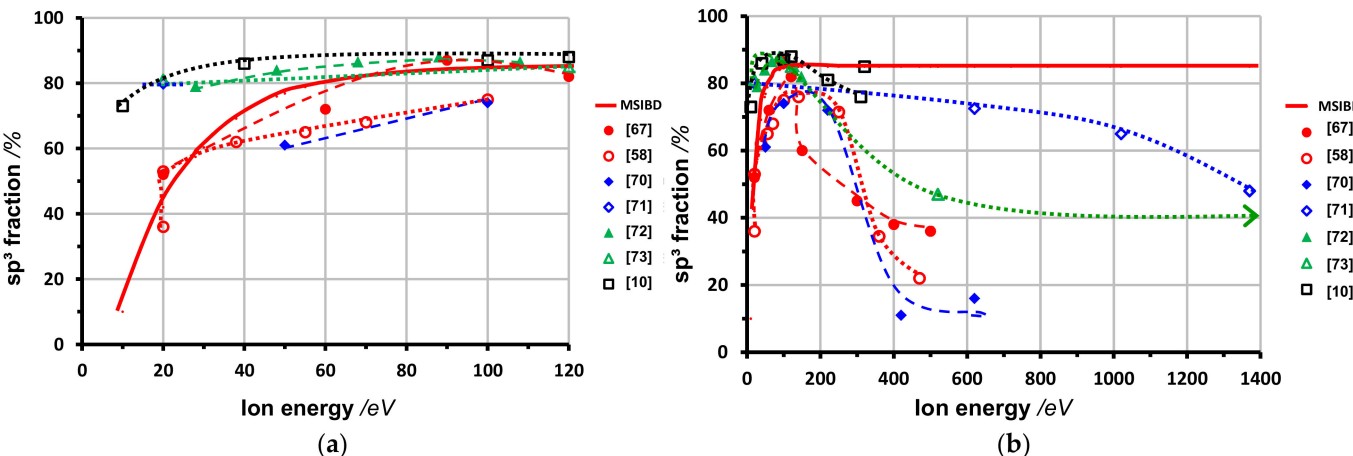

**Figure 10.** Dependence of the $sp^3$ fraction (from EELS) on the ion energy for carbon films, deposited by filtered vacuum arc. dc arc: [10,58,67,70–72], pulsed arc: [73] (300 A/5 ms, pulsed bias 2 µs on, 6 µs off). Empty symbols: ion energy for 0 V bias estimated by 20 eV. (**a**) Medium energy range 20–120 eV. Included Equation (12) for MSIBD (red line). (**b**) Extended energy range. Included is the curve for MSIBD ([43], red line).

In MSIBD, the high $sp^3$ fraction is retained for higher ion energies up to 300–400 eV (Figure 1b). Subsequently, the $sp^3$ fraction decreases slowly due to irradiation-enhanced relaxation processes. However, even with 1 keV ions, it amounts yet to 65% [43]. Moreover, in vacuum arc deposition, the $sp^3$ fraction remains sometimes on a high level above 40% up to 1 keV, but in most cases it decreases rather rapidly at much lower energies between 150 and 250 eV (Figure 10b). The difference to MSIBD can be explained (1) by the contribution of energetic ions from the high-energy tail and (2) by the much higher deposition rates and the correspondingly increased surface temperatures, which support the irradiation-induced relaxation.

The density shows a comparable dependence on the ion energy (Figure 11). It reflects the correlation between density and $sp^3$ fraction according to Equation (1) down to about

s ≈ 20% (Figure 12) (the lower densities of [73] could be caused by the used bias of 2 μs pulses with a duty cycle of 25%, resulting in a permanent alteration of low-energy and high-energy irradiation).

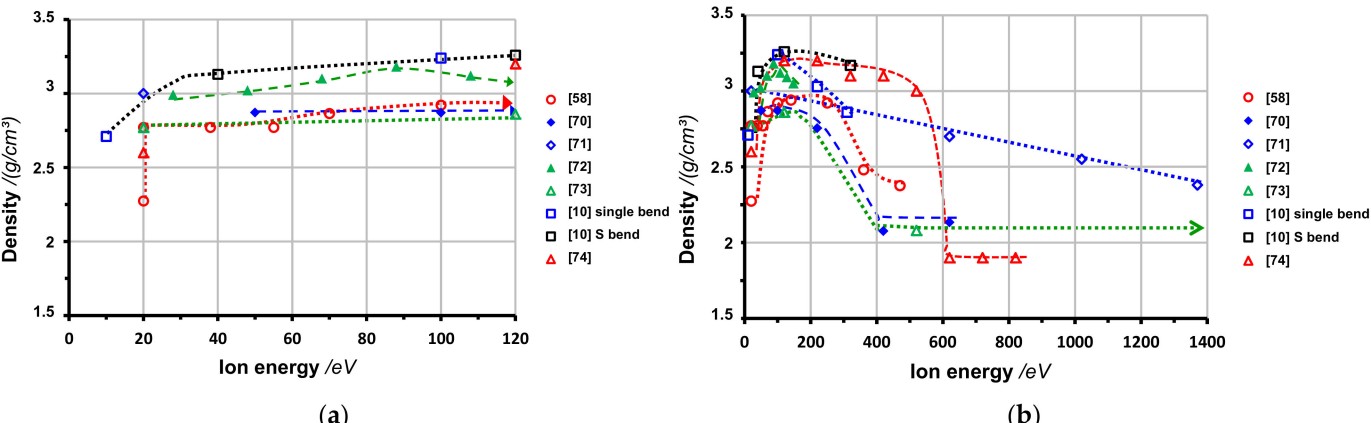

(**a**)          (**b**)

**Figure 11.** Dependence of the density on the ion energy for carbon films, deposited by filtered vacuum arc. (Density from plasmon peak according to Equation (5), [74]: from X-ray reflection (XRR)). dc arc: [10,58,70–72], pulsed arc: [73] (300 A/5 ms, pulsed bias 2 μs on, 6 μs off), [74] (1600 A/300 μs, synchronized bias). Empty symbols: Ion energy for 0 V bias estimated by 20 eV. (**a**) Medium energy range. (**b**) Extended energy range.

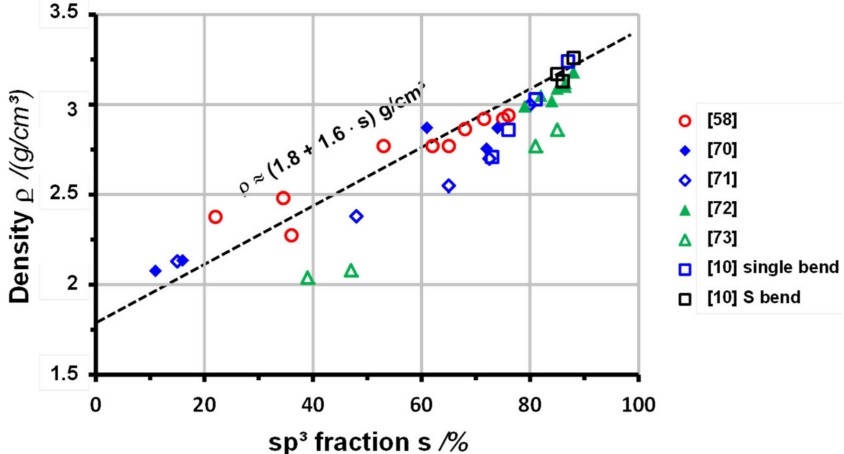

**Figure 12.** Relation of the density (derived from the plasmon energy according to Equation (5)) with the sp$^3$ fraction for carbon films, deposited by filtered dc arc at various ion energies. Data from Figures 10 and 11. Included pulsed arc (5 ms) with bias pulses of 2 μs with duty cycle of 25% [10,58,70–73]. The black dashed line corresponds to Equation (1), approximately valid down to near s = 20%.

The same holds also for the relation of the Young's modulus with density ρ and sp$^3$ fraction s over a broad range of ion energies (Figure 13): the directly determined values of the Young's modulus coincide very well with the values, estimated by Equation (3) from the density (determined by X-ray reflection (XRR)). In this case, the both properties (density and Young's modulus) decrease only weakly after the maximum at 100 eV and fall rapidly not before about 500 eV. This behavior differs markedly from the most results in Figures 10 and 11 (apart from [71]). It reflects the influence of the respective relaxation conditions, especially of the local surface temperature, depending on the specifics of the used deposition unit and deposition technology. For the investigations in Figure 13 a large industrial Laser-Arc machine with a two-fold rotating planetary and in the pulsed arc mode (duty cycle only 10%) has been used, allowing a more efficient heat dissipation and

corresponding­ly reduced surface heating in contrast to the usual laboratory devices with fixed substrates. The device-depending variations of the carbon structures at higher ion energies in Figures 10, 11 and 13 show, that the sp$^3$ maximum around about 100 eV in vacuum arc deposition does not result from the short-time relaxation in the thermal spike directly following the ion impact, as it was assumed in the early theories of Davies [75] and Robertson [76]. Rather, it is caused by the long-time relaxation under the influence of the specifically enhanced surface temperatures in combination with the perpetual ion impact.

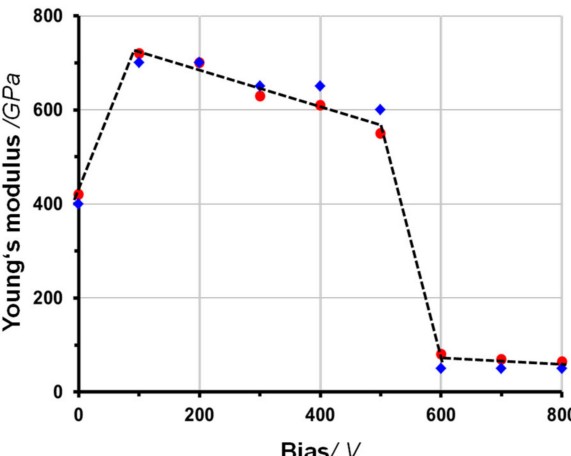

**Figure 13.** Dependence of the Young's modulus on the bias voltage for filtered Laser-Arc deposition with grounded anode (data from [74]). red circles: direct determination by laser acoustics, blue diamonds: estimation from the XRR density in Figure 11 with Equation (3).

The exclusive effect of irradiation-enhanced relaxation has been demonstrated in [77–79], using pulsed bias voltages of some kilovolts. Due to the short pulse length of 25–50 µs at a repetition frequency between 10–1000 Hz, corresponding to a duty cycle of only 820.03–5%, an essential influence on the mean surface temperature can be excluded. The films have been compared with the deposition on grounded substrates (bias 0 V) and under optimum conditions (bias 85 V, corresponding to ion energies around 100 eV). With increasing kilovolt bias, the films become more and more graphitic (Figure 14). The sp$^3$ fraction falls from about 85% to 30–40%. Asymmetry and hump of the Raman peak indicate an increasing Raman D peak, i.e., the formation of graphitic rings. Correspondingly, the Raman peak ratio $I_D/I_G$ increases from about 0.1 towards 1. The suppression of the Raman signal from the underlying silicon substrate at 960 cm$^{-1}$ hints to an essentially higher absorption. The formation of graphitic nanostructures leads to the formation of a granular surface topography with an enhanced roughness.

High deposition energies lead to overwhelmingly sp$^2$-bonded films with graphitic lamellae structures. As a response to the high compressive stresses, induced by the energetic ion impact, they are preferentially oriented perpendicular to the substrate surface, thus using the higher compressibility of the graphene stacks (Figure 15a). The formation of such oriented structures is accompanied by a fall of the through-film resistance by several orders of magnitude [80]. The heterogeneous structure due to separated graphitic regions leads to columnar growth at too low and at too high ion energies, which is clearly revealed at the fractured surface and by the roughness of the film surface (Figure 15b). This contrasts with the deposition at medium ion energies with dominating sp$^3$ networks, where top surfaces and fractured cross sections are smooth and structureless.

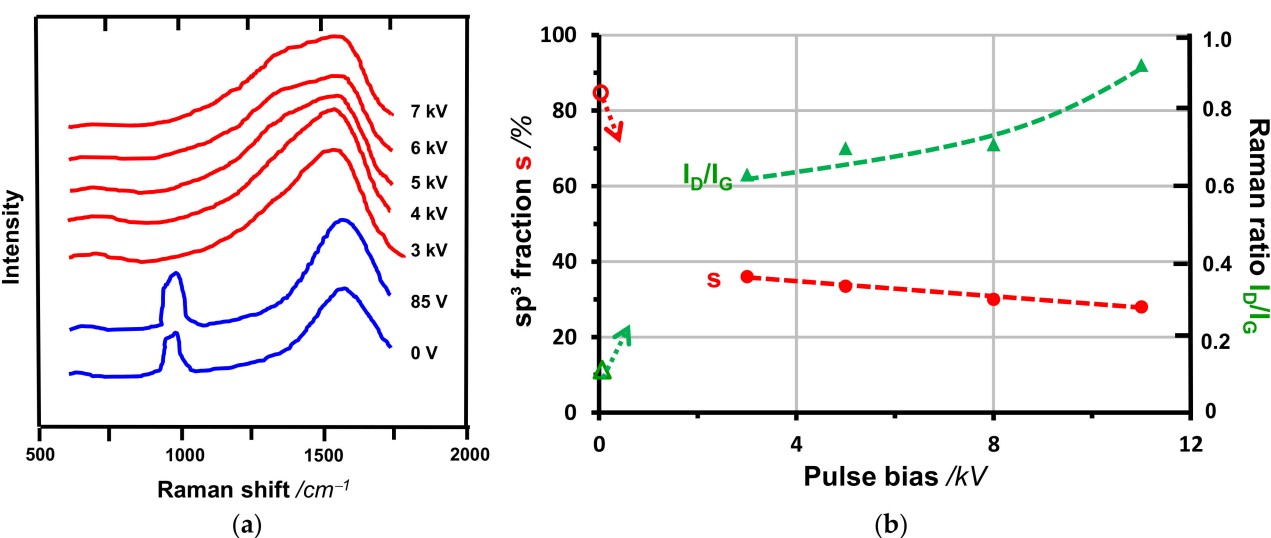

**Figure 14.** Raman investigation (excitation wavelength 514 nm) of carbon films, prepared by filtered dc vacuum arc deposition with various bias voltages. (**a**) Raman spectra for a dc bias of 0 and 85 V (blue lines) and for a pulsed bias (25 µs, 600 Hz, red lines). Data from [77]. (**b**) sp$^3$ fraction and Raman peak ratio for a dc bias of 85 V (empty symbols) or for a pulsed kV bias (25 µs, 600 Hz, solid symbols). Data from [79].

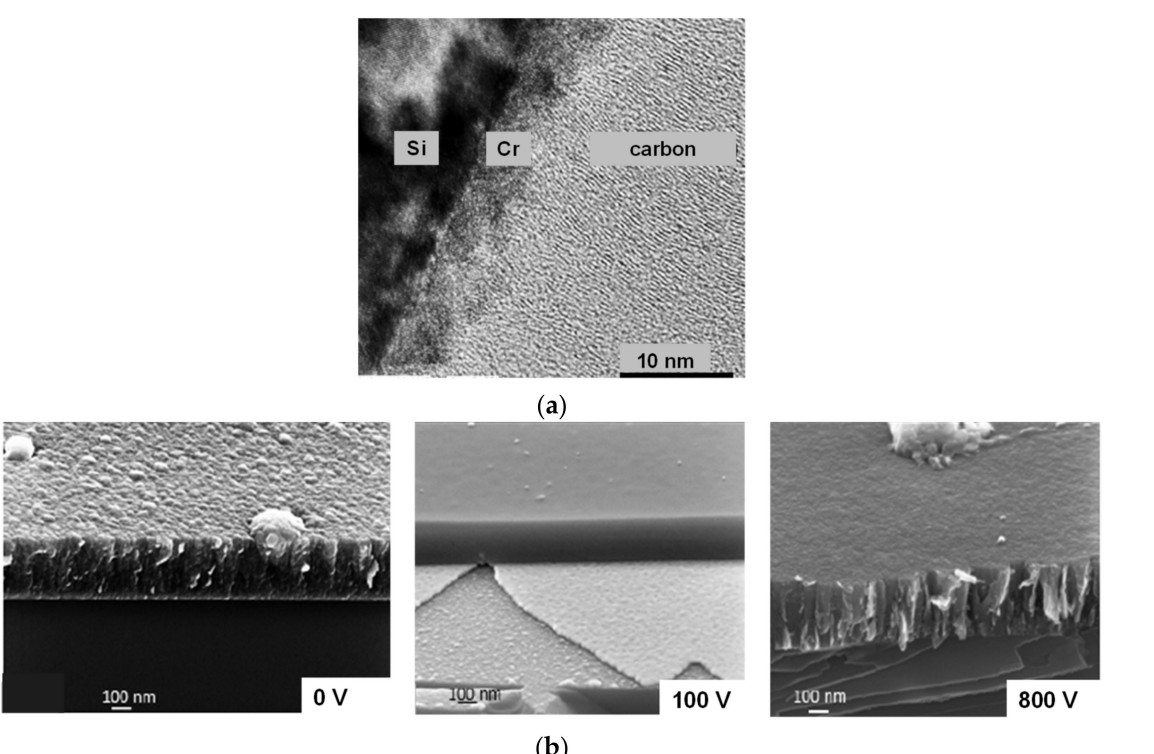

**Figure 15.** Carbon films, deposited by filtered anodic Laser-Arc (1.6 kA, 300 Hz) [74]. (**a**) Cross section, bias voltage 800 V (TEM image). (**b**) Cross section, bias voltage 0, 100, and 800 V (SEM image).

The partial relaxation of the very high compressive stresses (10–12 GPa in the maximum) starts already at ion energies of about 150 eV. Above 300 eV, they achieve a reduced, but still high stress level between 4 and 5 GPa up to ion energies of nearly 1 keV (Figure 16). Only with ions of some keV do the stresses decrease further to 1 GPa and below (Figure 17) [77,78]. The hardness reduces to about the half, but the compressive stresses falls much further, from about 10 GPa to below 1 GPa. This opens up the possibility

for efficient stress reduction (for improved adhesion) with only minor loss of hardness. Obviously, stress relaxations occur already with small atomic displacements, which do not essentially alter the connectivity of the carbon network. Apart from the lowered stress, the film adhesion is further markedly improved by the formation of a thicker intermixing layer due to the kilovolt pulses.

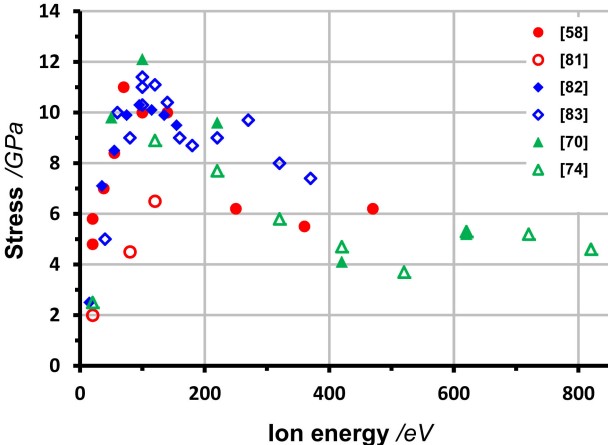

**Figure 16.** Dependence of the compressive stress on the ion energy for carbon films, deposited by filtered dc vacuum arc. Included values for filtered anodic Laser-Arc [58,70,74,81–83]. Empty symbols: ion energy ≈ 20 eV + e U$_{bias}$.

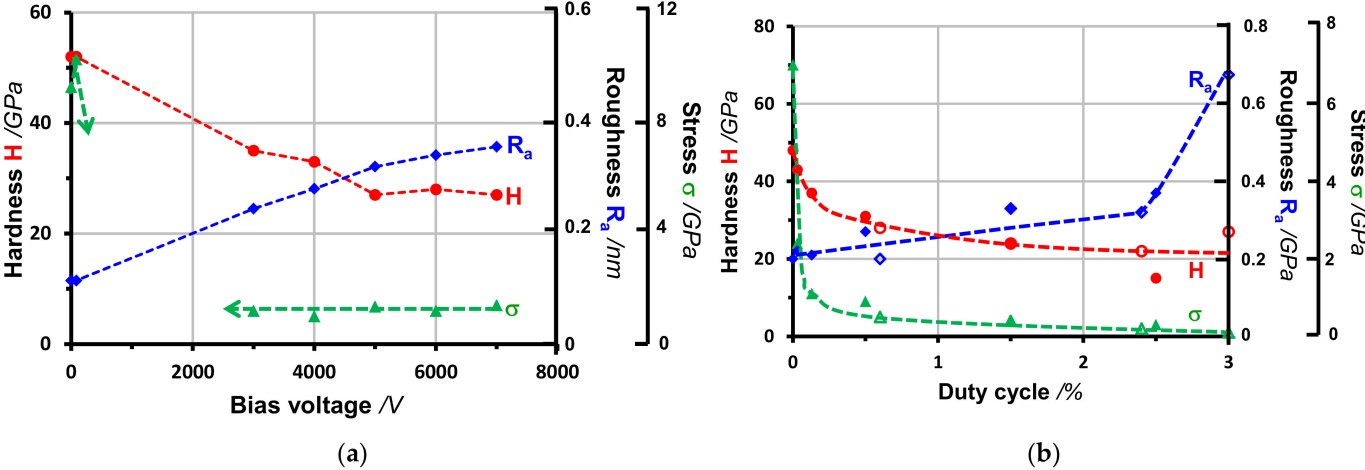

(**a**)  (**b**)

**Figure 17.** Hardness, compressive stress and roughness of carbon films, prepared by filtered vacuum arc deposition. (**a**) Dependence on the bias voltage (dc bias of 0 V and 85 V, pulsed bias of 3–7 kV with 25 µs, 600 Hz). Data from [77]. (**b**) Dependence on the duty cycle of 5 kV bias pulses. (Solid symbols: variation of the frequency of 25 µs pulses, empty symbols: variation of the pulse length at 600 Hz). Data from [78].

A decisive parameter for tribological applications is given by the hardness H. Hardness values in the superhard range above 40 GPa are sufficient to avoid abrasive wear by nearly any counterbody. According to Equations (2) and (4), they correspond to Young's moduli above 400 GPa and sp$^3$ fractions above 50%. With vacuum arc deposition at low temperatures, they are already realized with the mean "natural" ion energy of about 20 eV on unbiased substrates (Figure 10a). From 20 eV up to 50 eV, sp$^3$ fraction (and correspondingly Young's modulus and hardness) increase steeply and above 50 eV more slowly to the maximum values around 100 eV. However, for the wear, these further improvements of the already superhard films are usually of less importance.

From the technological point of view, depositions with lower ion energies are generally preferable: Due to the less energetic irradiation, they avoid the relaxation enhancement, reduce the thermal input and lower the formation of very large compressive stresses, which can induce delamination. Additionally, they demand a less elaborated power supply. However, there are two reasonable exceptions: (1) improved adhesion by the formation of a thin intermixing layer, deposited at the beginning with high energies and (2) the preparation of multilayer stacks of alternating harder and weaker layers by periodic variation of the bias voltage, to improve the toughness and to reduce the intrinsic stresses.

### 4.3. Deposition Temperature

For the deposition with (nearly) monoenergetic carbon ions (as with MSIBD), the transition from overwhelmingly $sp^3$-bonded to predominantly $sp^2$-bonded films occurs rather sharply around a critical temperature $T_c$ of about 150 °C for the optimum ion energy range around 100 eV (Figure 7a). The critical temperature increases slightly for higher ion energies. Carbon depositions with arc discharges principally shows a comparable behavior at enhanced temperatures. In the example of Figure 18, the structural and mechanical properties change markedly above 200 °C in a narrow range of less than 100 K [67,84]. Beyond this interval, they are nearly temperature-independent. Accordingly, three temperature ranges can be differentiated:

(1)    The low temperature range below $T_c$ with constant values of $sp^3$ fraction s, density $\rho$, Young's modulus E and stress $\sigma$. For varying ion energies, these quantities are related by Equations (1)–(4).

(2)    The transition range with strongly changing properties. The density decreases much more, and Young's modulus and stress much less than it would expected according to the reduced $sp^3$ fraction. Hence, the temperature-induced relaxation leads to other structures rather than the deposition with insufficient ion energies. The relations Equations (1)–(4) are not valid in the thermal transition range.

(3)    The high temperature range with nearly completely $sp^2$-bonded films and constant density and stress.

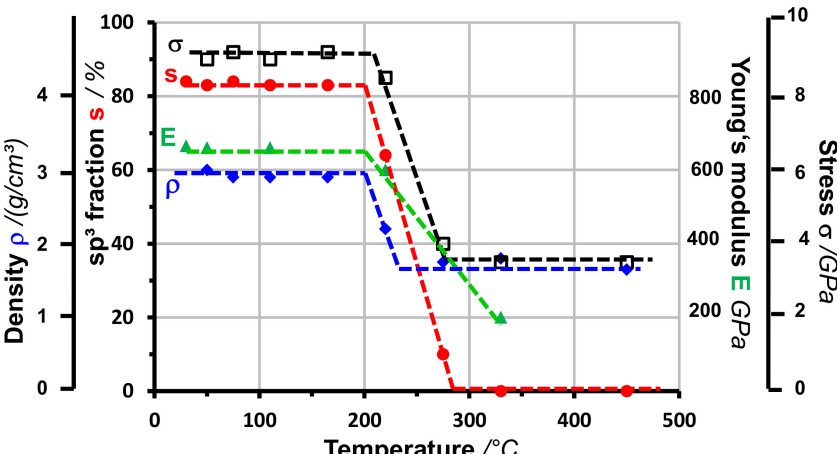

**Figure 18.** Dependence of structural and mechanical properties on the substrate temperature for carbon films prepared by vacuum arc deposition. Filtered dc arc deposition with ion energies of 90 eV. Data from [67,84].

However, several factors can influence the observed temperature dependence:

-    High mean deposition rates, especially in combination with high ion energies, can lead to increasing temperatures in the surface region. The real deposition temperatures are then above the measured bulk temperatures. Hence, the observed critical substrate temperatures lie below the real surface values. The deposition induced surface heating

is of especial importance for industrial technologies, where high deposition rates are usually an economic precondition.

- Instantaneous very high deposition rates, as they are possible with pulsed high current arcs, limit the interval for structural relaxation. If they are combined with low repetition frequencies, additional heating by the ion impact can be avoided. For momentary rates of 1000 nm/s, critical temperatures $T_c$ up to 400 °C have been realized [69].

- The carbon ions in vacuum arc discharges are distributed over a certain energy range (Figure 9). Correspondingly, the transition may occur more gradually and extend over a broader temperature range.

Figures 19–22 demonstrate the influence of additional factors. Notwithstanding the comparable high $sp^3$ contents ≥70% in the case of low-temperature deposition (typical for ion energies above 50 eV), the $sp^3$ fractions decrease either gradually over a broad temperature range or steeply in a narrow temperature interval (Figure 19). In the first case (observed for filtered arc deposition with ion energies of 60 and 80 eV), the decline starts already around 100 °C, but results at 300 °C only to a reduction to s ≈ 50% [85,86]. In the latter case (for filtered arc deposition with ion energies of 90 and 100 eV), the $sp^3$ fraction remains unchanged up to 200 °C and drops down at 300 °C to about 20% [70] or even to nearly zero [67,84]. The strange initial rise up to a $sp^3$ maximum at 140 °C in [70] has also been observed with 50 eV ions, but is not supported by the continuous slope of the density in Figure 20. It may be caused by temperature-induced release of unintentionally picked-up hydrogen.

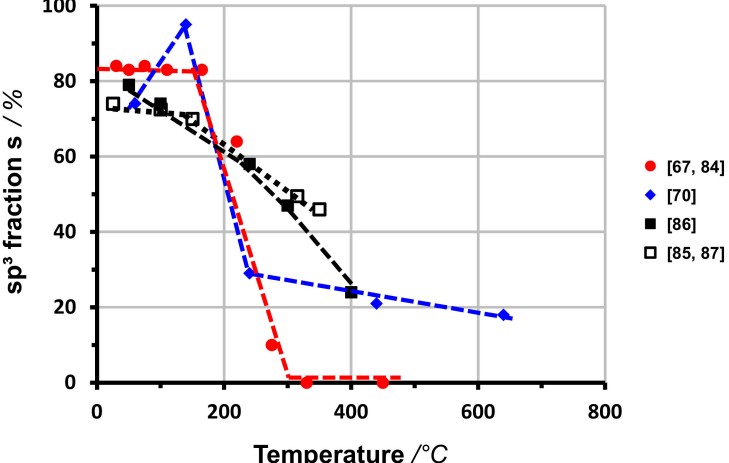

**Figure 19.** Dependence of the $sp^3$ fraction on the substrate temperature for carbon films prepared by filtered arc deposition with ion energies of 60 eV [85,87], of 80 eV [86], of 90 eV [67,84] and of 100 eV [70].

A possible explanation could be derived from the stress-induced-relaxation model [3] (Section 8.4) [68]: According to this model, the resulting $sp^3$ fraction is mainly determined by two processes: (1) the penetration of the impinging carbon ions with the generation of $sp^3$ bonds according to the local pressure and temperature conditions, (2) the stress-driven diffusion from the overstressed growth region towards the surface with stress relaxation and partial $sp^3 \rightarrow sp^2$ transformation. The diffusion is intensified by higher surface temperatures and by structural damaging due to energetic ion impact. At lower ion energies with accordingly smaller penetration depth, the shorter distance to the surface allows an essential stress relief already with few diffusion steps, i.e., at lower temperatures. At higher ion energies, the $sp^3$ decline occurs only at correspondingly higher temperatures. It is then supported by the increasing irradiation induced damaging of the superficial layer, leading to the observed steeper slope.

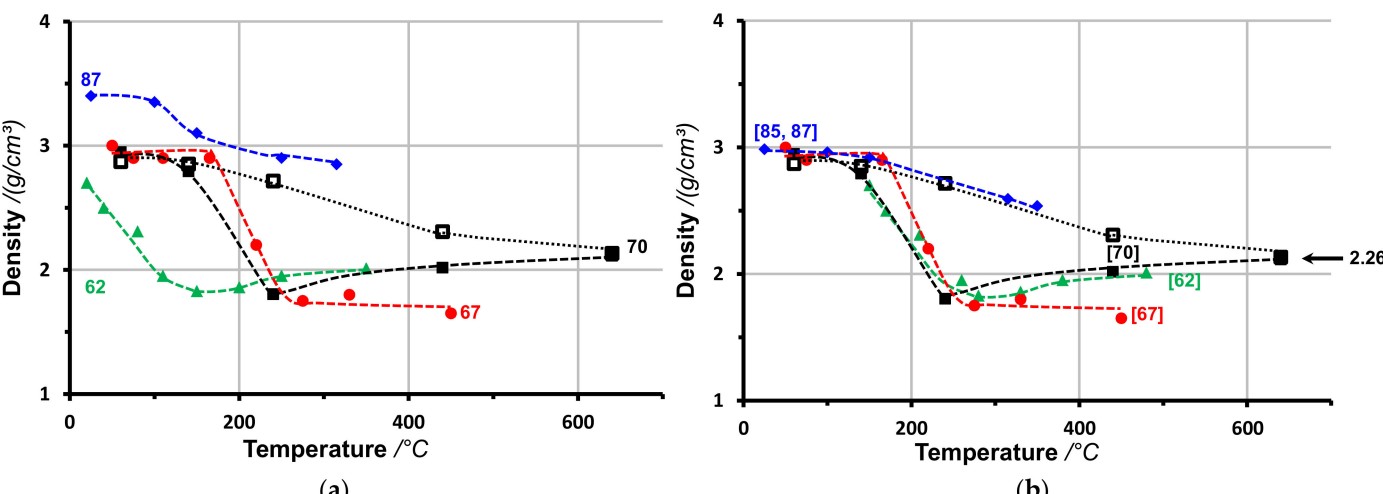

**Figure 20.** Dependence of the density on the substrate temperature for carbon films prepared by vacuum arc deposition. Filtered dc arc deposition with ion energies of 50 eV (empty black squares [70]), 60 eV (blue diamonds [87]), 90 eV (red circles [67]) and 100 eV (solid black squares [70] and pulsed cathodic Laser-Arc (80–90 eV, green triangles [62]). (**a**) Original values. (**b**) Corrected values: 1) ρ ([87]) from s ([85]) in Figure 18 according to Equation (1) (instead from XRR), ρ ([62]) shifted upwards by 130 K due to assumed ion induced surface heating.

The relations of the film density with the nominal deposition temperature show similar differences concerning the onset and the range of the density reduction (Figure 20). The deviations may partially be caused by a temporarily higher surface temperatures due to very high instantaneous fluxes of energetic ions as in the case of the Laser-Arc with high pulse currents around 1 kA [62]: Mean deposition rates of 2 nm/s, comparable to the filtered dc-arc deposition in [84], correspond at a pulse length of 100 μs and repetition rates of 200 Hz to a duty factor of 2%, leading to fiftyfold higher instantaneous ion flux rates. The ρ(T) relation from [62] in Figure 20a can be adapted to those from [67,70] by a tentative temperature shift of 130 K (Figure 20b). Deviating relations can also be caused by different characterization methods. The rather high densities from [87] are obtained from fitting the X-ray reflectivity under grazing incidence. However, the densities from the corresponding $sp^3$ fractions from Figure 19 with Equation (1) result in values, comparable to other investigations (Figure 20b). The densities from [62,67,70] have been determined by evaluation of the plasmon peak according to Equation (5). Now, the temperature dependence reveals two clearly separated modi: At not too high ion energies (50 eV [70], 60 eV [85,87]), the density decreases continuously to the graphitic values around 2 g/cm³ over a broad range from about 150 °C to above 500 °C. For filtered dc arcs at higher ion energies (90 eV [67], 100 eV [70]) and for cathodic Laser-Arc with 80–90 eV [62], the transition occurs in a small range of 100 K. This could again be explained by the enhancement of the thermal relaxation processes by the impact of high energy ions (the temperature dependence in MSIBD has been investigated alone for carbon ions with high energies above 90 eV. Hence, the gradual course of the pure thermal relaxation has not been observed). The irradiation leads also to the disturbance of the now mainly $sp^2$-bonded structure with densities clearly below 2.0 g/cm³, i.e., with a certain nanoporosity. Only at high temperatures are the atomic arrangements more ordered to graphitic structures and the density approaches the ideal graphite value $\rho_g$ = 2.26 g/cm³.

The comparison of Figures 19 and 20 shows that the density of temperature-relaxed structures in the transition region lie markedly below the values $\rho_{th}(s)$, calculated with Equation (1). It corresponds to a certain nanoporosity, developed as a by-product of the temperature-induced rearrangement. According to the density, the structural variations with increasing deposition temperature can be roughly classified into four groups (Table 4).

**Table 4.** Classification of carbon films, deposited by vacuum arc discharges at various temperatures.

| Temperature | Process | Density | Bond | Structure |
|---|---|---|---|---|
| low | subplantation | $\approx \rho_{th}(s)$ | $sp^2$, $sp^3$ | disordered $sp^2/sp^3$ mixture |
| enhanced | subplantation + partial bond relaxation | $< \rho_{th}(s)$ | $sp^2$, $sp^3$ | disordered $sp^2/sp^3$ mixture + $sp^2$ cluster, nanoporosity |
| high | complete bond relaxation | $< \rho_g$ | $\approx sp^2$ | disordered $sp^2$ + graphitic cluster, nanoporosity |
| very high | crystalline ordering | $\approx \rho_g$ | $sp^2$ | distorted graphitic structures |

The Young's modulus behaves similarly to the $sp^3$ fraction and the density (Figure 21). Sharp decline within an interval $\leq$100 K in the case of high ion energies above 90 eV, which are clearly indicated by the high Young's modulus around 700 GPa at low deposition temperatures [40,62,84]. In contrast, the Young's modulus decreases continuously from 50 to 450 °C for the medium energy deposition with 50 eV ions, as it is reflected by the low-temperature module of only 500 GPa [63]. The apparent shift of the observed transition temperatures to below 100 °C for the pulsed Laser-Arc depositions can again be explained by the additional transient heating due to the very high instantaneous deposition rates.

At low temperatures, the Young's modulus can be rather well estimated from the $sp^3$ fraction or the density according to Equations (2) and (3), as it is demonstrated in Figure 13 and for the [84] values in Figure 21. However, with the onset of thermal relaxation, the experimental Young's modulus lies markedly above the expected values. This means that the carbon network is less weakened than the reduced $sp^3$ fraction would suggest.

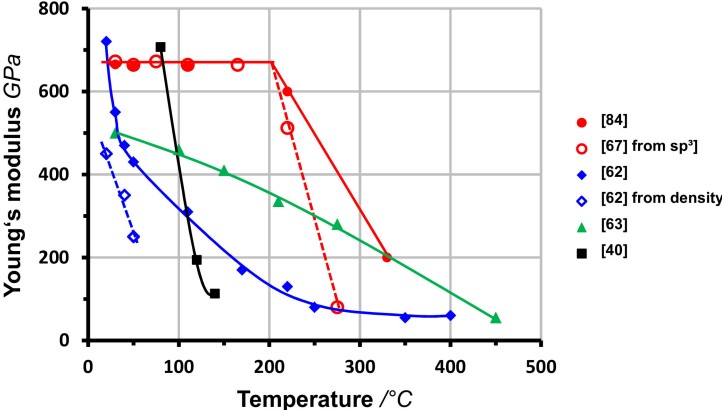

**Figure 21.** Dependence of the Young's modulus (determined by laser acoustics) on the substrate temperature for carbon films prepared by vacuum arc deposition. Filtered dc arc deposition (mean ion energy 90 eV [67,84]), filtered anodic Laser-Arc deposition (biased substrates, 87 eV [40]), and unfiltered cathodic Laser-Arc deposition (grounded substrates, 80–90 eV [62], 48 eV [63]). Empty symbols denote the estimations from the $sp^3$ content [67] and from the density [62] according to Equations (2) and (3).

The supporting effect of energetic ion irradiation on the thermal relaxation explains the different temperature behavior of carbon films, deposited by the Laser-Arc technique without and with curved filter sheets. By the filter unit, the ion energy is increased, leading to higher $sp^3$ fraction and higher Young's modulus at room temperature, but also to an irradiation-enhanced relaxation of the overstressed structure and to a correspondingly smaller transition range (Figure 22). The narrower energy distribution of filtered ion beams may additionally assist the sharper transition.

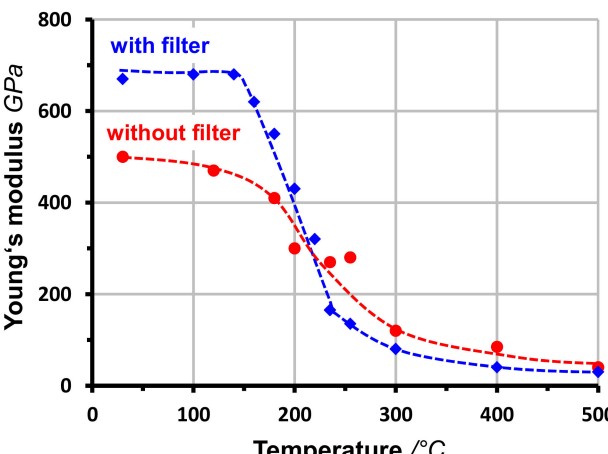

**Figure 22.** Dependence of the Young's modulus on the substrate temperature for carbon films prepared by Laser-Arc deposition without and with an additional filter unit (Fraunhofer-IWS).

The temperature sensitivity of the deposition process is of large technological importance: it determines the influence of possible temperature variations under industrial conditions (e.g., preceding cleaning procedure, batch filling, substrate material, substrate geometry, deposition duration etc.). For industrial applications, technological stability is an indispensable precondition, notwithstanding such changing conditions. Due to its lesser temperature sensitivity, the energy range around 60 eV (with sufficient high $sp^3$ fractions between 70% and 80% for low temperature deposition) should be more suitable than the range around 100 eV with maximum $sp^3$ contents of 80–85%. This holds especially for processes with high deposition rates, which tends towards additional surface heating.

The increasing graphitization for higher deposition temperatures is clearly revealed in the modification of the Raman spectrum: the increasing formation of aromatic rings and the corresponding rise of the D peak leads firstly to a shoulder and then to the emergence of a separated peak (Figure 23). For amorphous carbon films with $sp^3$ fractions near and above 70%, as they are realized by low temperature deposition with ion energies above about 60 eV, the Raman peak ratio $I_D/I_G$, the G peak position $\nu_G$ and the G peak width $\Delta\nu_G$ have typical values $I_D/I_G \approx 0.2$, $\nu_G \approx 1560 \text{ cm}^{-1}$ and $\Delta\nu_G \approx 200 \text{ cm}^{-1}$. For higher deposition temperatures, the Raman ratio increases, the G peak position shifts to higher wave numbers and the G peak width decreases.

In detail, the observed temperature dependence varies essentially for different investigations, even in the case of nominally comparable ion energies (Figure 24). The Raman peak ratio increases partially only to values between 0.5 and 1.5 at 400 °C [86,87], whereas in [62,88] already a value around 3 has been observed at a temperature of 150 °C. However, the different slope is in accordance with the deviating temperature dependence of the $sp^3$ fraction and the density (Figures 19 and 20). Only for the data from [62,88] are nearly completely $sp^2$ bonded structures ("nanostructured carbon") achieved in the investigated temperature range, whereas in [85–87], the film consists of a mixture of $sp^2$- and $sp^3$-bonded atoms with $sp^3$ fractions above 45% and 25%, respectively ("amorphous carbon"). Depending on the density, the Raman peak ratios lie around the same curve (Figure 25a). For the nanostructured range, it can be approximated by

$$I_D/I_G \approx 3.8 - 1.2\ \rho/(g/cm^3) \approx 1.6 - 1.9\ s, \qquad s \approx 0.84 - 0.52\ I_D/I_G \qquad (16)$$

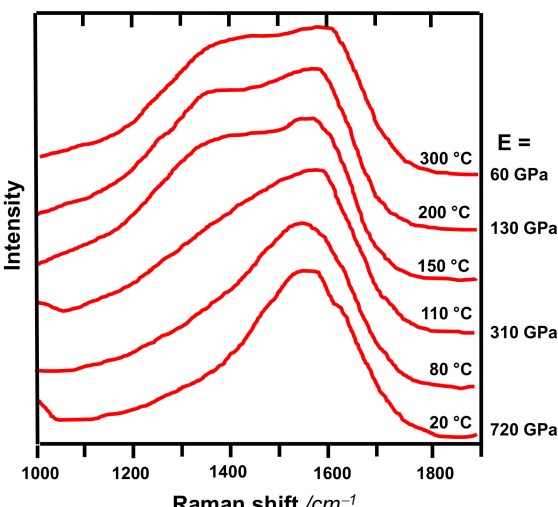

**Figure 23.** Raman spectrum of carbon films deposited by cathodic Laser-Arc without bias for various substrate temperatures [62].

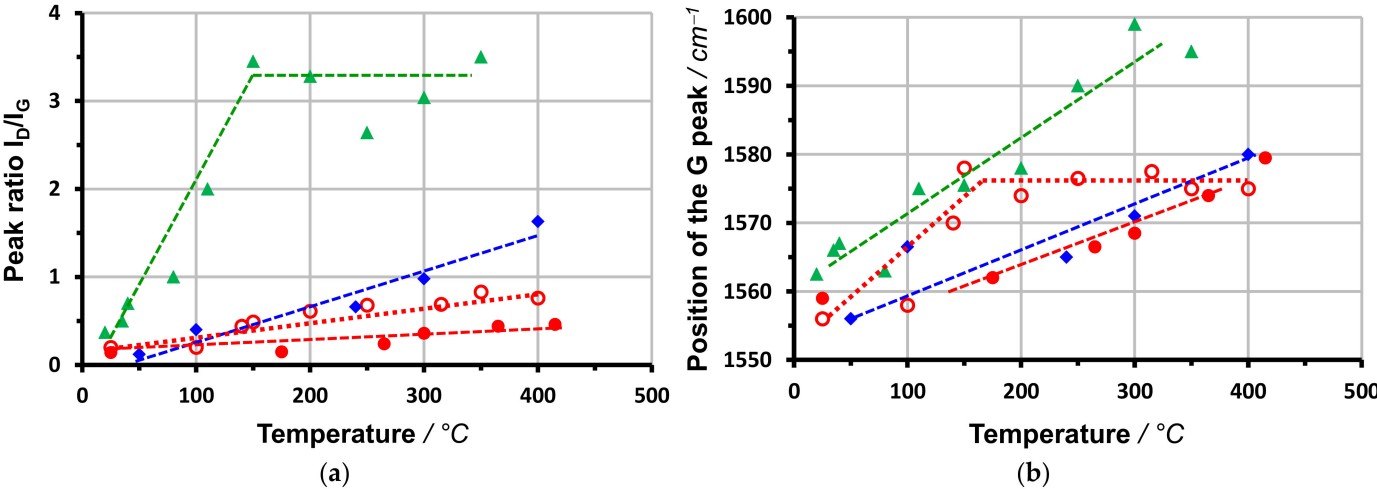

**Figure 24.** Variation of the Raman characteristics for carbon films in dependence on the deposition temperature. Data from [62,88] (green triangles, cathodic Laser-Arc deposition with ion energies of 80–90 eV), from [87] (red circles, filtered arc deposition with ion energies of 60 eV (empty symbols) and 100 eV (solid symbols) and from [86] (blue diamonds, filtered arc deposition with ion energies of 80 eV). (**a**) Raman peak ratio $I_D/I_G$. (**b**) Position of the G peak.

According to Equation (16), the boundaries for ta-C films ($s \geq 0.5$, $\rho \geq 2.6$ g/cm$^3$) are given by $I_D/I_G \approx 0.6$ and for amorphous films ($s \geq 0.2$, $\rho \geq 2.2$ g/cm$^3$) by $I_D/I_G \approx 1.2$. Thus, in this range, the Raman peak ratio can be used for a rough estimation of the sp$^3$ fraction. In comparison to the more reliable EELS method, it is non-destructive, local and does not demand tedious preparations. First of all, Raman investigations can be done on the real parts (even with rough surfaces and complex shapes), where the temperature conditions can deviate markedly from the reference silicon slices for EELS.

In contrast, the relation of the G peak position with the density deviates markedly for the various investigations (Figure 25b). Correspondingly, a clear correlation between the both Raman characteristics $I_D/I_G$ and $\nu_G$ exists only for a particular investigation, whereby the slopes considerably differ (Figure 26). The differing behavior may be caused by additional factors, as the intrinsic stress, which may influence the position of the G peak, and as the dimensions of the graphitic clusters, which affect the Raman peak ratio. This means that the emerging arrangement of the sp$^2$-bonded atoms is essentially influenced by the specific relaxation conditions.

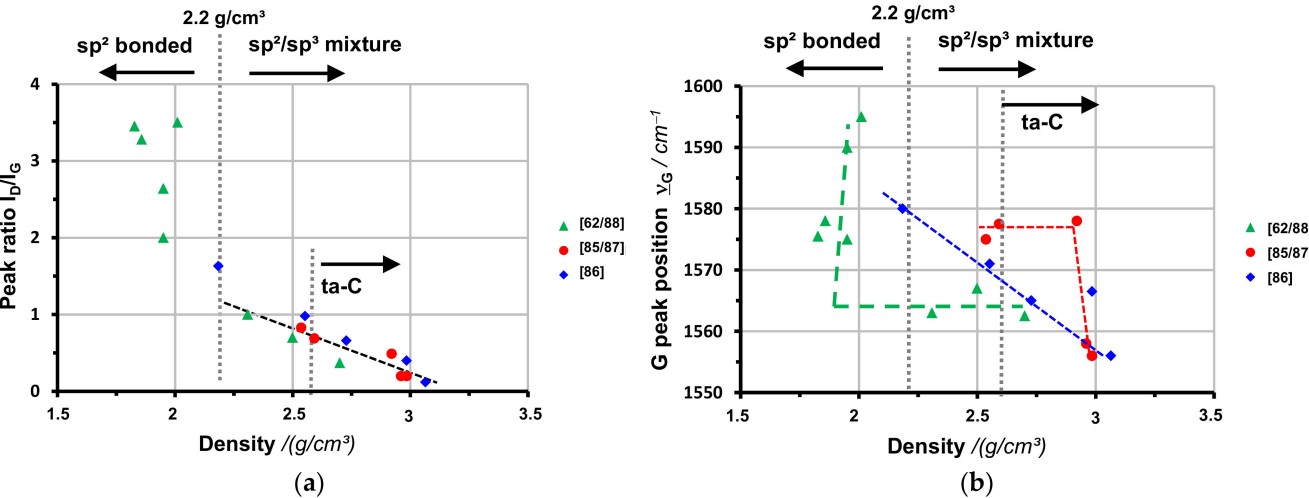

(**a**)　　　　　　　　　　　　　　　　　(**b**)

**Figure 25.** Relationship between the Raman characteristics and the density for carbon films deposited at different temperatures. Data from [62,88] (Laser-Arc deposition with ion energies of 80–90 eV), from [85,87] (filtered arc deposition with ion energies of 60 eV) and from [86] (filtered arc deposition with ion energies of 80 eV). Density for [85,87] calculated from the $sp^3$ fraction in Figure 18 with Equation (1). (**a**) Raman peak ratio. (**b**) G peak position.

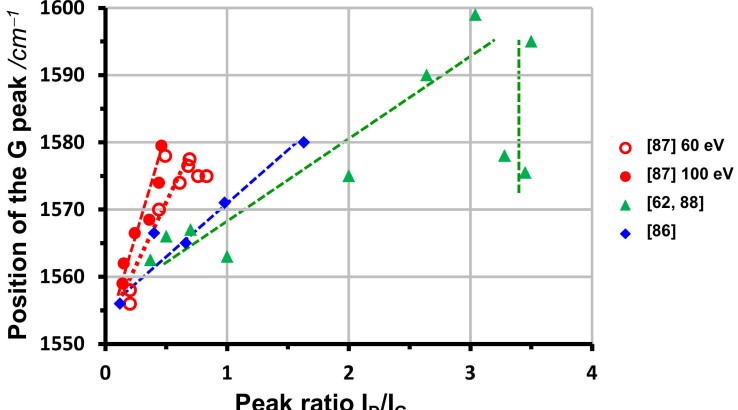

**Figure 26.** Variation of the G peak position for carbon films in dependence on the peak ratio $I_D/I_G$. Data from [62,88] (Laser-Arc deposition with ion energies of 80–90 eV), from [87] (filtered arc deposition with ion energies of 60 and 100 eV) and from [86] (filtered arc deposition with ion energies of 80 eV).

　　　The structural relaxation is driven by the reduction of the stored elastic energy, i.e., of the high compressive stresses. Thus it leads to a preferred orientation of the c-axis parallel to the surface [89]. Due to the weak interaction between neighbored graphene layers, they have a much higher compressibility in this direction in comparison to the high stiffness of the covalent bonds within the layer. For carbon layers, deposited by vacuum arcs above around 200 °C, the formation of the curved graphene ribbons perpendicular to the surface has been clearly proved by TEM investigations and by electron diffraction [40,70,90–94]. The perpendicular orientation occurs also at inclined incidence of the carbon ions under 45° or 65° [46] or 70° [95]. Hence, collisional effects such as selective sputtering are not responsible for the observed structural anisotropy. The formation of an anisotropic structure at higher temperatures is also revealed by the increasing divergence of the Young's modulus values, determined by indentation and by laser acoustics, which probe the stiffness mainly in normal and in lateral direction [40].

### 5. Structure Map

Figure 27 summarizes the effect of ion energy and deposition temperature on the structure of arc deposited carbon films. Due to the limited experimental data and due to the influence of the specific deposition conditions, the boundaries (especially at higher energies and temperatures) represent only the qualitative tendencies, the given energies and temperatures are only typical values. It generalizes the structural phase diagram, developed in [70].

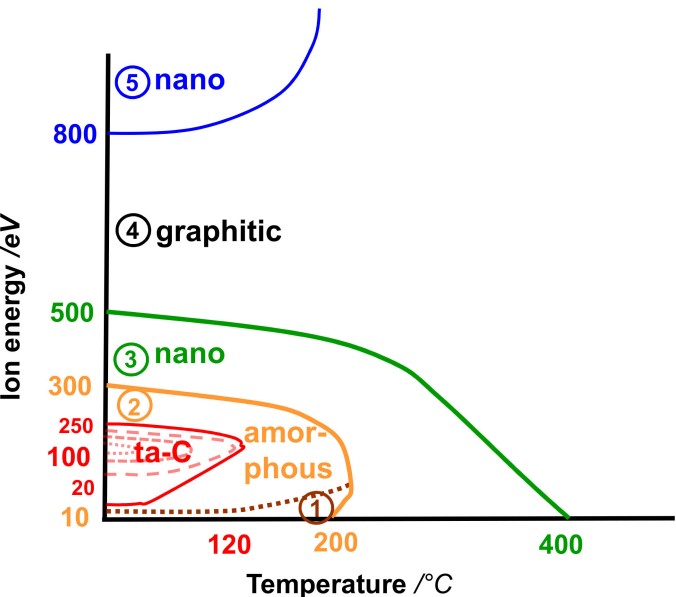

**Figure 27.** Hypothetical structural phase diagram of vacuum arc deposited carbon films. The temperature and energy values denote only typical values.

Roughly, the following regions can be distinguished:

(1) Low ion energy deposition ($\leq$10 eV)

Such conditions are typical for evaporation and also for sputtering with minor ion assistance. In the arc deposition, the energy range below 20 eV is only realized by ion scattering in inert atmospheres, not discussed here. At such low ion energies, the subplantation mechanism does not work. The carbon films grow by the usual condensation on top of the surface. The films are usually completely $sp^2$-bonded. Sometimes, minor contents of carbynelike $sp^1$ bonds have been observed [96]. At not too high deposition temperatures (typically below 200 °C), the impinging carbon atoms have only a limited surface mobility. Correspondingly, the films are amorphous and have a marked nanoporosity.

At enhanced temperatures ($\approx$200–300 °C), the higher surface mobility leads to the formation of hexagons and their coalescence to larger clusters, indicated by the increasing roughness. The film consists of a mixture of disordered and ordered regions with some porosity ("nanostructured carbon"). According to evaporation results, these processes start around 200 °C [97].

At high temperatures ($\geq$300 °C), the film consists completely of distorted graphitic structures. In sputter deposition at higher deposition temperatures, the structural relaxation over only short distances leads to an increasing porosity up to above 30% [98,99]. The strong anisotropy of the electrical conductivity suggests a preferential arrangement of the graphitic layers parallel to the substrate surface [100].

(2) Medium ion energy deposition (typically between 10 and 300 eV)

The subplantation mechanism leads to the formation of $sp^3$ bonds. In combination with low temperatures, ion energies between 20 and 250 eV allow the formation of ta-C films with $sp^3$ fractions above 50%. The highest values are achieved with energies around

100 eV. With increasing temperature and correspondingly intensified thermal relaxation, the structure changes to more and more sp$^2$-bonded structures: from amorphous carbon with a mixture of disordered sp$^2$- and sp$^3$-bonded atoms to nanostructured carbon of sp$^2$-bonded atoms in a composite of disordered atoms and graphitic clusters and finally to graphitic carbon with distorted stacks of graphene layers.

For ion energies around the ta-C range (typically 10–20 and 250–300 eV), the films are overwhelmingly sp$^2$ -bonded. For not too high temperatures, they remain amorphous with an atomic mixture of sp$^3$- and sp$^2$-bonded atoms.

(3)    High ion energy deposition (typically between 300 and 500 eV)

At higher ion energies of some hundred electronvolts, the temperature-induced relaxation is more and more supported by the irradiation enhanced mobility. The sp$^2$-bonded atoms agglomerate to graphitic nanoclusters ("nanostructured carbon"). Increased ion energy and increased temperature lead to comparable structures. The properties are more and more determined by the frequency, size and mutual arrangement of the graphitic clusters, partially reflected by the Raman characteristics.

(4)    Very high ion energy deposition (typically between 500 and 800 eV)

At further enhanced ion energies and/or substrate temperatures, the film consists completely of sp$^2$ bonded atoms. They arrange in stacks of distorted graphene layers ("graphitic carbon"). Under the influence of the high compressive stresses, they are preferentially oriented perpendicular to the surface.

(5)    Ultrahigh ion energy deposition (typically above 800 eV)

At even higher ion energies, the formation of such extended structures is hindered by the competition of impact-induced destruction and temperature-driven relaxation. The size of the ordered regions decreases. The film consists of a nanostructured mixture of very small and strongly distorted graphitic clusters and amorphous regions of sp$^2$-bonded atoms. At such ultrahigh ion energies, the formation of graphitic film structures (with distinct graphene layer stacks) is only possible at correspondingly enhanced deposition temperatures.

## 6. Conclusions

With their "natural" ion energies of about 20 eV, vacuum arc depositions of carbon films yield already on unbiased substrates sp$^3$ fractions near 50%, resulting in superhard hardness values around 40 GPa, sufficient for most tribological applications. The films consist of a disordered mixture of sp$^3$- and sp$^2$-bonded atoms with only a minority in graphitic rings ("amorphous carbon"). The sp$^3$ fraction increases rapidly from near 50% to the maximum values some above 80% for ion energies in the optimum range around 100 eV. Higher ion energies lead to heating and damaging of the surface layer, resulting in irradiation-enhanced thermal relaxation. The structure changes to overwhelmingly sp$^2$-bonded arrangements with an increasing fraction of graphitic clusters ("nanostructured carbon") and further to distorted graphitic structures. Due to the high compressive stresses, the graphene layers are preferentially oriented perpendicular to the surface. By further increased ion energy, the formation of ordered regions is more and more supressed, leading again to composites of graphitic and amorphous regions of the sp$^2$-bonded atoms.

At enhanced temperatures, the structures are modified under the combined effect of thermal activation and ion impact. Both factors act partially in the same direction and partially contrarian. Medium ion energies promote the formation of sp$^3$ bonds in competition with thermal relaxation. However, at higher energies, the sp$^3$-to-sp$^2$ transformation and also the formation of graphitic regions are supported by the irradiation-induced damage and the correspondingly increased mobility in the surface region. However, ultrahigh ion energies demolish the surface structures, thus counteracting the thermal ordering processes.

Generally, the combined effect of these decisive factors, ion energy and instantaneous local temperature, seems to depend sensitively on their transient coincidence, which is influenced by the specific conditions, especially the instantaneous deposition rate.

The qualitative picture, sketched above, has been extracted from various investigations of vacuum arc deposition of carbon films. Usually, they report similar trends, but in detail they show often noticeable deviations, depending on the specific experimental device. Partially, these differences may be caused by different measuring and data evaluation methods. However, the main problem consists of deviating relations between the given external deposition conditions (bias voltage, mean substrate temperature) and the decisive internal parameters for the film growth, especially the ion energy (inclusive its distribution) and the surface temperature (inclusive its temporal variation).

Another aspect concerns the technological robustness of the industrial film preparation, i.e., the consequences of unintentional deviations from the selected optimum conditions. An especially critical point is the temperature rise in the aimed high-rate deposition. Surface heating and damage can be reduced by using ion energies as low as possible. However, in the low-energy range, the film structure is strongly influenced by energetic variations, complicating the reproducibility. In the deposition of complex shaped parts, inclined incidence is inevitable. This has a similar effect as an impact with reduced ion energy and leads to corresponding loss in the film hardness. In industrial coaters, the parts are usually assembled on planetaries, where they rotate around one, two or three axes and experience (in the mean) the same conditions. Due to the periodically changing impact angles, the films get a multi-layered structure with alternating hard layers from nearly normal incidence and less hard layers from more grazing incidence, whereas most characterization methods record the mean values only.

Thus, the comparison of different investigations shows comparable trends for the film structure and properties with the controllable external parameters. However, due to the complex conditions in the arc deposition, the quantitative relations are influenced by the specific coating device and the used coating technology. This complicates the transfer of technological experience to modified devices or modified batches. Hence, a significant and practicable characterization of the produced films is all the more important. In vacuum arc deposition for tribological applications with the precondition of high hardness, the $sp^3$ fraction lies clearly above 30%. In this range, $sp^3$ fraction, hardness, Young's modulus, density and Raman peak ratio are rather strongly correlated (Equations (1)–(4), (16)). The selection depends mainly on the practicability and the availability of the chosen method. Due to their direct influence of the tribological behavior, the film characterization by means of the mechanical properties (using indentation or laser acoustics) is recommended. For local measurements, the Raman peak ratio may be used. For films with a lower $sp^3$ fraction, as they are arc-deposited only under rather unusual conditions (high temperatures, very high ion energies, inert gas), the situation is less sufficient: The $sp^3$ fraction is only of minor importance and the density is mainly determined by the porosity. The Raman characteristics reflect sensitively the arrangement of the $sp^2$-bonded atoms. Thus, they can be used for the control of the reproducibility. However, their relation to the properties of tribological relevance is not clear. Hence also in this case, the direct characterization by means of the mechanical properties is recommended.

**Funding:** This research received no external funding.

**Institutional Review Board Statement:** Not applicable.

**Informed Consent Statement:** Not applicable.

**Data Availability Statement:** Data sharing is not applicable to this article.

**Conflicts of Interest:** The author declares no conflict of interest.

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
