# Peer review of "Structure and Characterization of Vacuum Arc Deposited Carbon Films—A Critical Overview"

_coatings, doi:10.3390/coatings12020109_

Round 1

Reviewer 1 Report

Carbon films with excellent properties have been widely used in many fields, and can be deposited by various methods, such as evaporation, ion beam sputtering, evaporation, ion beam sputtering, pulsed laser deposition, vacuum arc. This review comprehensively summarize the research about structure and characterization of vacuum arc deposited carbon films. The sp³ fraction, hardness, Young’s modulus, density and Raman peak ratio are rather strongly correlated. This work is of significance to cabon films and its application. I think it can be accepted after minor revision.

1, The determination of sp3 fraction by XPS or EELS, The quantitative relation between sp3 fraction and Raman spectra, are not detailly described in this review.

2, More constructive ideas and prospect about carbon films need to be provided.

Author Response

Thank you very much for your critical revision and the resulting hints.

 “1, The determination of sp3 fraction by XPS or EELS, The quantitative relation between sp3 fraction and Raman spectra, are not detailly described in this review.”

The Raman spectrum reflects the arrangement of the sp² bonds and is only indirectly influenced by the sp³ bonds. Hence, the same sp³ fraction can result in different Raman parameters (“hysteresis” in Ferrari-Robertson, Phys. Rev B 61, 14095 (2000), compare also Figs. 14 and 25). A universal quantitative relation Raman – sp³ cannot be given and I am skeptical concerning the derivation of sp³ fractions from the Raman spectrum. Nevertheless, the Raman ratio reflects within a series of comparable experiments the qualitative sp³ tendency (see Figs. 14 and 25) and represents thus a valuable tool for the technological optimization.

“2, More constructive ideas and prospect about carbon films need to be provided.”

The topic of this overview consists in summing up the relations between deposition conditions on one hand and structural parameters and mechanical properties on the other hand with some recommendations for suitable deposition ranges and for significant characteristics. The film development according to specific (tribological) applications and their technological realization will be discussed in other contributions of this issue and is outside the scope of this overview.

Reviewer 2 Report

The article can be published as it is. It is well written and structured. The plots are clear and easy to follow.

 However, the paper could be further improved by addressing the following minor issues:

1) Revision of the following sentence should be considered:
"For the usual excitation with visible light, the Raman spectrum of carbon materials is (due to resonance enhancement) completely determined by the sp² bonded carbon atoms."

 This does not seem to be factually accurate. For example, nanocrystalline diamond is also a carbon material but the Raman spectrum is not completely determined by the sp² bonded carbon atoms. It is usually dominated by the sp3 carbon peak. Sp2 carbon is also present but does not determine the appearance of the spectrum.

2) Some terms should be explained for a non-specialist audience.
The author uses terms such as “natural” ion energy, “natural” kinetic energy. The term “natural” is sometimes used in the literature on vacuum arc deposition, but it may not be clear to some readers what the term means. A short explanation should be provided.

3) Sources cited in this manuscript are quite old. Half of them are over 20 years old. Less than 10% of cited sources are younger than 5 years. The author should consider adding several recent publications related to vacuum arc deposition.

Author Response

Bernd Schultrich

Structure and characterization of vacuum arc deposited carbon films – A critical overview

Reviewer 2

Thank you very much for your critical revision and the resulting hints.

1) Revision of the following sentence should be considered:
"For the usual excitation with visible light, the Raman spectrum of carbon materials is (due to resonance enhancement) completely determined by the sp² bonded carbon atoms."
This does not seem to be factually accurate. For example, nanocrystalline diamond is also a carbon material but the Raman spectrum is not completely determined by the sp² bonded carbon atoms. It is usually dominated by the sp3 carbon peak. Sp2 carbon is also present but does not determine the appearance of the spectrum.

In the introduction, the field of carbon materials under consideration is now explicitely restricted to pure carbon films, prepared by PVD methods, thus excluding the CVD diamond films,
page 1 now: “Thin film deposition by PVD methods allows a broad combination, especially of sp² and sp³ bonded configurations, (nominally) without other elements as hydrogen. “

page 4 now: “For the usual excitation with visible light, the Raman spectrum of PVD carbon films is (due to resonance enhancement) completely determined by the sp² bonded carbon atoms, notwithstanding a fraction of up to 90 % sp³ bonds in disordered arrangements. (The diamond peak at 1322 cm-1 occurs only for sp³ bonds in crystalline diamond grains.)”

 “2) Some terms should be explained for a non-specialist audience.
The author uses terms such as “natural” ion energy, “natural” kinetic energy. The term “natural” is sometimes used in the literature on vacuum arc deposition, but it may not be clear to some readers what the term means. A short explanation should be provided.

page 13 now: “For usual arc currents below 1 kA, the carbon ions propagate in the plasma with a mean velocity of about 18 000 m/s, corresponding to a “natural” kinetic energy e0 » 20 eV within the arc plasma [54, 55].”

3) Sources cited in this manuscript are quite old. Half of them are over 20 years old. Less than 10% of cited sources are younger than 5 years. The author should consider adding several recent publications related to vacuum arc deposition.

The reference list reflects the development of the scientific activities (and not my permanent personal literature research activities) in this field. The highest intensity in fundamental studies on the deposition of pure carbon films was indeed concentrated between 1995 and 2005, notwithstanding the remaining open questions, addressed in this review. In the following years, the main interests shift more and more to application-oriented problems and to new carbon materials (as alloyed carbon films, graphene, ..). I am always very interested in any hints to supplementary literature.  

Reviewer 3 Report

The article “Structure and characterization of vacuum arc deposited carbon films – A critical overview” represents survey research and is devoted to the analysis of the influence of deposition conditions on the formation of the structure and properties of hydrogen-free carbon coatings.

The results provided in this article correspond to the aim and scope of the journal “Coatings”.

The paper shows the formation features of carbon coatings deposited by vacuum arc methods, i.e., the effect of ion energy and substrate temperature on the structure and properties of carbon coatings.

It is worth noting that hydrogen-free carbon coatings, produced via arc methods, are a highly topical issue of modern materials science.

After reviewing the manuscript, I would recommend “Accept after minor revision”

Please, consider the following comments:

  1. The article, submitted to the journal at the end of 2021, represents survey research, but the literature references are of 20-30 years ago. For your information, only 5 articles out of 99 sources have been published over the past 5 years.
  2. The article lists: "... graphitic, nanostructured and amorphous films ...". However, the author did not mention Linear-Chain Nanostructured Carbon (carbyne films) - the structure of these carbon-based coatings is significantly different from the listed above.
  3. The article lacks keys to the abbreviations, e.g., to PVD, IBAD, PLD, etc. Please, provide them to all the abbreviations used in the paper, namely when they are first mentioned. (page 2 line 57, 61).
  4. Page 3 line 112 «…For a ta-C foil of circa 50 nm thickness, …..» what is meant by “foil”?
  5. Page 4 line «a very small escape depth». This statement is not entirely accurate - using the XPS method, it is possible to perform profiling and determine the distribution of carbon bonds over the thickness of the coating.
  6. Sections 2.3. Raman spectroscopy and 2.3. Young’s modulus / hardness have the same number 2.3 (page 4 line 125 and page 5 line 203)
  7. Section «2.3. Raman spectroscopy» - in my view, when analyzing the La values, it is necessary to take into account the results given in the article “Ferrari, A. C., & Robertson, J. (2000). Interpretation of Raman spectra of disordered and amorphous carbon. Physical Review B, 61(20), 14095–14107. doi:10.1103/physrevb.61.14095", in particular, consider the results of this article to justify the choice of the TK equation (Tuinstra and Koenig) for determining the La cluster size

Tuinstra, F. Raman spectrum of graphite / F. Tuinstra [et al.] // The Journal of Physical Chemistry. – 1970. – Vol. 53. – P. 1126–1130.

  1. «3. The reference case: Ion beam deposition» - it is not entirely correct to compare the structures of carbon coatings obtained by the MSIBD method and vacuum arc methods. The MSIBD method is characterized by the complete absence of macroparticles in the flow and monoenergetic flow of carbon ions, while arc methods are characterized by a certain number of macroparticles in the plasma flow, the number and size of which are determined by the energy modes of the evaporator operation, as well as a fairly wide energy spectrum (DOI: 10.1016/B978-012513910-6/50051-7)

It is also worth noting that these macroparticles are the fragments of a graphite cathode and when they hit the substrate, they are fixed on it, which leads to a decrease in the quality of the coating structure and an increase in the graphite component. (see the works by A. Anders, A.A. Voevodin, I.I. Aksenov, V.E. Strel'nitskij, A.V. Rogachev)

  1. The caption to figure 2 is separately on the next page (page 8 line 272, 273)
  2. I would recommend the author to specify what vacuum-arc methods can be used to obtain the energies of the ions shown in Figure 27, as well as to supplement each section 1-5 with this information. (pages 28, 29 line 874-922)
  3. In my opinion the conclusion is too long. It can be shortened to point out the key findings in these studies.
  4. Extraneous full stops need to be removed/deleted (page 33 line 1144, 1157)

Author Response

Bernd Schultrich

Structure and characterization of vacuum arc deposited carbon films – A critical overview

Reviewer 3

Thank you very much for your very detailed critical revision and the resulting hints.

1. “The article, submitted to the journal at the end of 2021, represents survey research, but the literature references are of 20-30 years ago. For your information, only 5 articles out of 99 sources have been published over the past 5 years.

The reference list reflects the development of the scientific activities (and not my permanent personal literature research activities) in this field. The highest intensity in fundamental studies on the deposition of pure carbon films was indeed concentrated between 1995 and 2005, notwithstanding the remaining open questions, addressed in this review. In the following years, the main interests shift more and more to application-oriented problems and to new carbon materials (as alloyed carbon films, graphene, ..). I am always very interested in any hints to supplementary literature.  

2. The article lists: "... graphitic, nanostructured and amorphous films ...". However, the author did not mention Linear-Chain Nanostructured Carbon (carbyne films) - the structure of these carbon-based coatings is significantly different from the listed above.

page 1 now: “Physical vapor deposition (PVD) allows the preparation of pure carbon films (nominally) without any other elements as hydrogen and of structures with a broad combination of the various bonding types, especially in sp² and sp³ bonded arrangements. (Carbynelike sp1 configurations are only realized by low-energy deposition, e.g. via gas-phase condensation, which is not considered in this overview.)”

page 33 now: “1) Low ion energy deposition (£ 10 eV)
Such conditions are typical for evaporation and also for sputtering with minor ion assistance. In arc deposition, the energy range below 20 eV is only realized by ion scattering in inert atmospheres, not discussed here. At such low ion energies, the subplantation mechanism does not work. The carbon films grow by the usual condensation on top of the surface. The films are usually completely sp² bonded. Sometimes minor contents of carbynelike sp1 bonds have been observed [96].”

“96.  Piedade, A.P.; Cangueiro, L. Influence of carbyne content on the mechanical performance of nanothick amorphous carbon coatings. Nanomaterials 2020, 10, 780. “

3. The article lacks keys to the abbreviations, e.g., to PVD, IBAD, PLD, etc. Please, provide them to all the abbreviations used in the paper, namely when they are first mentioned. (page 2 line 57, 61).

has been corrected

4. Page 3 line 112 «…For a ta-C foil of circa 50 nm thickness, …..» what is meant by “foil”?

page 4 now: “Hence, the necessary preparation of sufficient thin foils (by dissolving the silicon substrate), the used measuring technique and deviating evaluation procedures induce some uncertainty of the extracted sp³ values.” 

5. Page 4 line «a very small escape depth». This statement is not entirely accurate - using the XPS method, it is possible to perform profiling and determine the distribution of carbon bonds over the thickness of the coating.

Depth profiling demands the successive thinning of the film with the danger of structural modifications in the superficial region. Hence it is not recommended for XPS and AES investigations of diamondlike films.

6. Sections 2.3. Raman spectroscopy and 2.3. Young’s modulus / hardness have the same number 2.3 (page 4 line 125 and page 5 line 203)

o.K., has been corrected

7. Section «2.3. Raman spectroscopy» - in my view, when analyzing the La values, it is necessary to take into account the results given in the article “Ferrari, A. C., & Robertson, J. (2000). Interpretation of Raman spectra of disordered and amorphous carbon. Physical Review B, 61(20), 14095–14107. doi:10.1103/physrevb.61.14095", in particular, consider the results of this article to justify the choice of the TK equation (Tuinstra and Koenig) for determining the La cluster size
Tuinstra, F. Raman spectrum of graphite / F. Tuinstra [et al.] // The Journal of Physical Chemistry. – 1970. – Vol. 53. – P. 1126–1130.

Both references have been already taken into account (references [1, 26]). Because they use different evaluation procedures of the Raman spectrum (symmetric Lorentz and asymmetric BWF distribution vs. symmetric Gaussians), their quantitative results are only conditionally comparable.

8. «3. The reference case: Ion beam deposition» - it is not entirely correct to compare the structures of carbon coatings obtained by the MSIBD method and vacuum arc methods. The MSIBD method is characterized by the complete absence of macroparticles in the flow and monoenergetic flow of carbon ions, while arc methods are characterized by a certain number of macroparticles in the plasma flow, the number and size of which are determined by the energy modes of the evaporator operation, as well as a fairly wide energy spectrum (DOI: 10.1016/B978-012513910-6/50051-7)
It is also worth noting that these macroparticles are the fragments of a graphite cathode and when they hit the substrate, they are fixed on it, which leads to a decrease in the quality of the coating structure and an increase in the graphite component. (see the works by A. Anders, A.A. Voevodin, I.I. Aksenov, V.E. Strel'nitskij, A.V. Rogachev)

Indeed the macroparticles are a hard problem for the vacuum arc deposition of carbon films. Hence, they have been widely eliminated in all the investigations, discussed in this review. The cited investigations used the magnetic filtering with their nearly complete separation. We used the Laser-Arc technique (pulsed arc with laser ignition and controlled displacement of the arc spot) with drastic reduction of the macroparticle emission, partially supplemented by a filter unit. But also without an additional filter, highly tetrahedrally bonded films can be realized (see e.g. Fig. 21 with Young’s modulus above 700 GPa, corresponding to sp³ fractions above 80 %). Usually the very small impinging cathode particles represent only the nuclei for much larger growth defects, especially for thicker films. Hence, the film structure itself is less influenced. The main problem consists in the induced roughness and the possible detachment of the conelike growth defects, which can be critical for tribological applications.

“9. The caption to figure 2 is separately on the next page (page 8 line 272, 273)”

o.K., has been corrected

10. I would recommend the author to specify what vacuum-arc methods can be used to obtain the energies of the ions shown in Figure 27, as well as to supplement each section 1-5 with this information. (pages 28, 29 line 874-922)”

Because the carbon plasma in vacuum arc discharges is completely ionized, you can realize any ion energy by using suitable bias voltages. According to Fig. 9, the “natural” ion energy is usually above 10 eV. Principally, also the range below 10 eV could be realized with a sufficient countervoltage. But this case has not been systematically investigated, because only films with minor quality could be expected and this range is easier accessible by sputtering or evaporation.

11. In my opinion the conclusion is too long. It can be shortened to point out the key findings in these studies.”

I do not agree

12. Extraneous full stops need to be removed/deleted (page 33 line 1144, 1157)

o.K., has been corrected
